# Multiscale lineament analysis and permeability heterogeneity of fractured crystalline basement blocks

Alberto Ceccato*, Giulia Tartaglia, Marco Antonellini, Giulio Viola

Dipartimento di Scienze Biologiche, Geologiche ed Ambientali – BiGeA, Alma Mater Studiorum – Università di Bologna – via Zamboni, 67, 40126 Bologna, Italy

*Present address: Geological Institute, Department of Earth Sciences, ETH Zurich, Sonneggstrasse 5, 8092 Zurich, Switzerland

*Correspondence to*: Alberto Ceccato (aceccato@erdw.ethz.ch)

**Abstract.** The multiscale analysis of fracture patterns helps defining the geometric scaling laws and the relationships between outcrop- and regional-scale structures in a fracture network. Here, we present a novel analytical and statistical workflow to analyze the geometrical and spatial organization properties of the Rolvsnes granodiorite lineament (fracture) network in the crystalline basement of southwestern Norway (Bømlo Island). The network shows a scale-invariant spatial distribution described by a fractal dimension D ≈ 1.51, with lineament lengths distributed following a general scaling power-law (exponent □ = 1.88). However, orientation-dependent analyses show that the identified sets vary their relative abundance and spatial organization with scale, defining a hierarchical network. Lineament length, density, and intensity distributions of each set follow power-law scaling laws characterized by their own exponents. Thus, our multiscale, orientation-dependent statistical approach can aid in the identification of the hierarchical structure of the fracture network, quantifying the spatial heterogeneity of lineament sets and their related regional- vs. local-scale relevance. These results, integrated with field petrophysical analyses of fracture lineaments, can effectively improve the detail and accuracy of permeability prediction of heterogeneously fractured media. Our results show also how the geological and geometrical properties of the fracture network and analytical biases affect the results of multiscale analyses and how they must be critically assessed before extrapolating the conclusions to any other similar case study of fractured crystalline basement blocks.

## 1 Introduction

Crystalline rocks are characterized by very low intrinsic permeability, usually in the order of 10-18 m2 (Achtziger-Zupančič et al., 2017; Brace, 1984), so that that their capability to transmit and/or store fluids is mainly related to the structural permeability associated with fracture and fault networks created by brittle deformation and the associated fluid-rock interaction (e.g., Caine et al., 1996; Caine and Tomusiak, 2003; Ceccato et al., 2021a, 2021b; Evans et al., 1997; Pennacchioni et al., 2016; Schneeberger et al., 2018; Stober and Bucher, 2015). When studied at different scales, fracture and fault networks commonly exhibit variable geometrical and spatial characteristics, which may significantly affect the

overall permeability structure (spatial heterogeneity and anisotropy of permeable zones) of the fractured crystalline rock (Le Garzic et al., 2011; Hardebol et al., 2015; Holdsworth et al., 2019; Torabi et al., 2018). One way to obtain quantitative constraints upon the scale-dependency of fracture and fault network attributes is to perform a multiscale analysis of, for example, their length and spacing distributions. The aim of these multiscale analyses is to obtain scaling laws that can quantify the variability of fracture network properties across scales (Bonnet et al., 2001; Bossennec et al., 2021; Castaing et al., 1996; Chabani et al., 2021; Dichiarante et al., 2020; Gillespie et al., 1993; McCaffrey et al., 2020; Odling, 1997).

### 1.1 Presentation of the problem

Quantifications across scales and scaling laws are usually derived from the analysis of lineament maps traced by remote sensing techniques (e.g., Bour et al., 2002; Castaing et al., 1996; Odling, 1997). In the past, much effort has been invested in the definition of a direct mathematical, quantitative relationship between lineament map parameters and permeability/porosity parameters (Davy et al., 2006). However, in order to provide realistic qualitative and quantitative constraints on fracture network permeability, the analysis of lineament maps requires the integration of deterministic field inputs on the geology of the remotely sensed "lineaments" (e.g., Bertrand et al., 2015; Bossennec et al., 2021, 2022; Hardebol et al., 2015). Furthermore, lineament maps provide very large and statistically robust datasets, which are, however, subject to analytical, methodological, and interpretative biases (Peacock et al., 2019; Scheiber et al., 2015). Therefore, accurate statistical analyses need to be performed to evaluate the possible biases affecting each dataset and the extrapolation limits of the observations retrieved from their analyses (e.g., Bistacchi et al., 2020; Dichiarante et al., 2020; McCaffrey et al., 2020).

### 1.2 Structure of the paper

In this paper, we present a methodological approach, which, when informed by statistical tests, aims to support the decision-making process during the analysis and identification of the most appropriate scaling laws describing fracture network properties variability across scales, as derived from the analysis of multiscale lineament maps. In addition, we integrate previously obtained geological, geochronological and petrophysical data about the geology and permeability of the identified lineaments to constrain the permeability structure of fractured crystalline basement units.

We describe the case study of the Bømlo crystalline basement formed by the Rolvsnes granodiorite (Western Norway; Scheiber and Viola, 2018). Fracture network maps thereof were obtained from the manual picking of lineaments on LiDAR digital terrain models, aerial, and Unmanned Aerial Vehicle (UAV, drone) orthophotos of the exposure area of the Rolvsnes granodiorite (Fig. 1).

The analyzed attributes include: (i) fractal dimension $D$ of the lineament network; (ii) lineament orientation; (iii) cumulative length distribution of lineaments at each analyzed scale and for each orientation set; (iv) intensity/density scaling for the whole lineament network and for each orientation set; (v) heterogeneity of lineament spacing distribution. These parameters are usually adopted for the quantification of the spatial occupancy and of the "fractal" character of fracture-lineament

networks (fractal dimension $D$ and length distribution scaling laws), to constrain their physical connectivity and spatial organization (spacing and length), and, ultimately, for the quantification of the permeability structure of the host fractured medium (e.g. Bonnet et al., 2001; Healy et al., 2017; Nyberg et al., 2018; Peacock and Sanderson, 2018). The statistical tests adopted to constrain the most representative fitting curves and distributions include: (i) Least Square Regression (LSR) and Maximum Likelihood Estimation (MLE) coupled with Kolmogorov-Smirnov (KS) statistical tests adopted on cumulative distributions (Dichiarante et al., 2020; Kolyukhin and Torabi, 2013; Rizzo et al., 2017); (ii) bivariate box and whiskers plots to evaluate the distribution of fracture spacing heterogeneity parameters and their statistical significance.

Our statistical, orientation-dependent multiscale analysis permits to: (i) identify groups of regional- vs. local-scale lineament sets based on the variation of geometrical and scaling parameters; (ii) define statistically robust scaling laws for the geometrical properties and the range of scales within which those laws can be applied; (iii) evaluate the difference between scaling laws retrieved from the entire network and those from individual sets. The implications of the adoption of general scaling laws on the upscaling/downscaling of fracture network properties, as well as the possible analytical biases and sources of errors in the analytical approach, are then evaluated and discussed.

By integrating this information with existing field structural analyses and modeling of lineaments petrophysical properties (Ceccato et al., 2021a, 2021b, Scheiber and Viola, 2018), we provide further constraints on the multi-scale heterogeneity in magnitude, orientation, and spatial distribution of the permeability structure of the fractured crystalline basement.

The lineament network in the Rolvsnes granodiorite developed during a prolonged brittle tectonic history within massive, isotropic granitoid rocks. The in-situ analysis and characterization of the structural constituents of this fracture and fault zone network, e.g., the Goddo Fault Zone (GFZ, Fig. 1), has previously allowed us to reconstruct in detail a detailed timing of deformation and to quantify the geometry and petrophysical properties of the associated fractures (Ceccato et al., 2021a, 2021b; Scheiber and Viola, 2018; Viola et al., 2016), although the larger-scale geometry and organization of the fracture and fault network on Bømlo remained poorly constrained and thus needed further quantification.

## 2. Geological setting

The crystalline basement of the Island of Bømlo belongs to the Upper Allochthon units of the Caledonian orogen (Gee et al., 2008). Our lineament maps represent the fracture pattern affecting the Rolvsnes granodiorite, a pre-Scandian (466 ± 3 Ma; zircon U-Pb dating) granitoid pluton hosted in the Upper Allochthon metamorphic units (Scheiber et al., 2016) (Fig. 1). The Rolvsnes granodiorite recorded a prolonged and multi-phase brittle deformation history (Scheiber et al., 2016; Scheiber and Viola, 2018), only briefly summarized in the following, while the reader is referred to the cited literature for a more detailed and comprehensive description of the tectonic history of the area. Overall, the whole tectonic history of the area is the expression of three main deformation episodes (Bell et al., 2014; Fossen et al., 2016, 2021): (1) Caledonian convergence and continental collision from the Mid-Ordovician to the Silurian; (2) extensional tectonics related to the late-Scandian orogenic collapse during the Devonian, and (3) prolonged and multi-phase extensional tectonics related to the North Sea rifting from

the Permian to the Cretaceous. During this tectonic evolution, the pre-Scandian Rolvsnes granodiorite did not record penetrative ductile strain and was instead affected by pervasive brittle deformation. Each tectonic stage recorded in the granodiorite is associated with a characteristic set of fracture and fault zones that dissect Bømlo (Scheiber and Viola, 2018): (1) NNW- and WNW-striking conjugate strike-slip faults developed coevally with ENE-WSW and NE-SW-striking reverse faults during Caledonian convergence; (2) the same structures were reactivated with opposite kinematics during the early stages of late-Scandian orogenic collapse; (3) NW- and NNW-striking normal faults ascribable to the regional Permian-to-Jurassic rifting phase of the North Sea, which partially reactivated earlier, inherited structures. During the latest rifting stages of the North Sea, in the Early Cretaceous, new N- to NNE-striking fracture corridors and normal faults overprinted the previously formed fracture pattern. This tectonic history is reflected in the field by a sequence of three main classes of fracture and fault zones: (i) pre-Permian, ESE-WNW and NE-SW striking mineralized shear fractures and minor faults; (ii) Permo-Jurassic major normal faults, mainly NW-SE and N-S striking; (iii) Cretaceous fracture clusters striking N-S to NNE-SSW (Scheiber et al., 2016; Scheiber and Viola, 2018).

A key structure for the detailed analysis of the timing of deformation, the geometry of the deformation structures, and the effects of deformation on the petrophysical properties of the crystalline basement of Bømlo, is the Goddo Fault Zone (GFZ, Fig. 1), (Ceccato et al., 2021a, 2021b; Scheiber and Viola, 2018; Viola et al., 2016). The GFZ is an east-dipping normal fault that accommodated multiple slip increments during the prolonged Permian-to-Cretaceous rifting of the North Sea, recording several stages of reactivation, during which a complex network of brittle structural facies developed in the fault core (*sensu* Tartaglia et al., 2020). Structures like the GFZ actually controlled the permeability and fluid flow evolution from rifting to current times of the crystalline basement (Ceccato et al., 2021a, 2021b; Viola et al., 2016).

Similarly to Bømlo, a complex fracture and fault network affects the crystalline basement of the Utsira High, of which the Rolvsnes granodiorite is interpreted as the onshore analogue (Fredin et al., 2017; Trice et al., 2019). The Utsira High crystalline basement is composed of (likely multiple) pre-Scandian igneous intrusions of similar age and composition to the Rolvsnes granodiorite (Lundmark et al., 2014; Slagstad et al., 2011). The fracture network in the Utsira High developed under tectonic conditions like those of the Bømlo crystalline basement, but with several significant differences mainly related to the structural position of the two crystalline basements within the North Sea rifting region (Bell et al., 2014).

## 3. Materials and Methods

In this study, the word “lineament” is used to refer to any linear feature of the topography as detected on a digital representation of the surface. The topography of Bømlo is cut through by deep linear grooves resulting from the penetration at depth of erosional processes exploiting fracture and fault zones of the crystalline basement (Scheiber and Viola, 2018). Therefore, the mapped lineaments represent fracture and fault zones identified in the field. In the followings, the terms fracture(s) and fault zone(s) refers only to the geological structures observed in the field, which indeed assemble the remotely sensed lineaments. The lineament maps (Fig. 2a) used for the presented multiscale analyses have been generated in

ArcGIS 10.8 by manually picking the same digital terrain model (DTM) of selected areas of the Island of Bømlo at different scales of observation. DTM's from high-resolution (1 m/pxl) airborne Light Detection and Ranging (LiDAR) surveys (Fig. 1) have been used for the manual picking of lineaments at the 1:5,000; 1:25,000 and 1:100,000 scales. The details of LiDAR data acquisition and DTMs elaboration can be found in Scheiber et al. (2015). In addition, the dataset of lineaments interpreted from LiDAR DTM at the 1:5,000 scale was integrated with the interpretation of aerial orthophotos from the Bing Maps database (https://www.bing.com/maps). Bing aerial imagery was also adopted to distinguish between natural and man-made linear structures and to check for artefacts and potential misinterpretation of linear features on LiDAR-derived DTMs in the absence of systematic ground truthing. The 1:100 outcrop-scale lineament picking was performed on digital orthophotos of a key GFZ outcrop (Figs. 1, 3a) as obtained from the elaboration of the imagery collected via UAV-drone surveys through Structure-from-Motion (SfM) algorithms. Details on this acquisition and its elaboration methods can be found in Ceccato et al. (2021a). Topographic lineaments were traced as single, linear segments (not polylines) interpreting their topographic expression on DTMs. The obtained lineament maps are included in the dataset related to the present paper available at https://data.mendeley.com/datasets/4zdjpmr9jk/1. This interpretation technique introduces two major analytical biases on the obtained lineament maps: 1) the interpreted length may only partially represent the entire lineament (which may be covered by deposits or be differently expressed in the topography, thus being not visible in its entire length, e.g., Cao and Lei, 2018); 2) as a consequence, abutting relationships, intersections between lineaments and lineament network topology and connectivity remain highly speculative and susceptible to subjective biases (Andrews et al., 2019). The orientation of mapped lineaments, expressed as azimuth angle from the geographic north, was calculated in ArcGIS 10.8 using Easy Calculate 10 (https://www.ian-ko.com/free/free_arcgis.htm) and the Orientation Analysis Tools (https://is.muni.cz/www/lenka.koc/prvnistrana.html). Rose diagrams plotting lineament azimuths were produced with the MARD 1.0 software (Munro and Blenkinsop, 2012). Lineament density $P_{20}$ ($m^{-2}$) and intensity $P_{21}$ ($m/m^2$) (Dershowitz and Herda, 1992) were calculated as the ratio between the total number of lineaments and total length of lineaments, respectively, over the total area of the land exposure in each lineament map.

### 3.1. Fractal dimension – Box-counting method

The fractal dimension of each lineament map at different scales was computed with the box-counting method (Bonnet et al., 2001; Gillespie et al., 1993) by using the freely available function boxcount.m in MATLAB R2019b (http://www.fast.u-psud.fr/~moisy/ml/boxcount/html/demo.html). The box-counting method consists in subdividing the analyzed image in progressively smaller square boxes of side $b$ and counting how many of them contain a segment of the analyzed lineament network. Plotting the number of boxes $N_b$ containing at least one lineament against the side length $b$ on a log-log diagram should yield a straight curve, whose slope defines a power-law function with $D$ as the fractal exponent (Bonnet et al., 2001). The fractal dimension obtained from the box-counting method quantifies the scaling properties of the spatial occupancy of the lineament network (Bonnet et al., 2001).

### 3.2. Cumulative length distribution analyses

Length data of lineaments have been organized as cumulative distributions and plotted in log-log diagrams of the length $L$ of lineaments on the X-axis versus $N_{(l>L)}$, the cumulative number of lineaments with length $l > L$ (Fig. 2b-c). The cumulative length distributions were then normalized by the area of the land surface reported on each map over which the lineaments were picked. Single-scale distributions report the lineament lengths at a specific scale of observation. Multiscale distributions report the sum of the cumulative length distributions observed at different scales. We have analyzed the multiscale/single-scale normalized distribution functions of (i) all lineaments included in each lineament map at different scales and (ii) each lineament orientation set at different scales.

#### 3.2.1. Fitting of multiscale cumulative length distributions

The single-scale cumulative length distributions have been merged to form a single multiscale distribution. The mathematical functions fitting multiscale cumulative distributions are commonly retrieved by manual fitting of the distributions by assessing the slope of the tangent to the observed distributions (e.g., Bertrand et al., 2015; Bossennec et al., 2021; Castaing et al., 1996; Chabani et al., 2021; Le Garzic et al., 2011). Here we apply a more quantitative method by adopting least square regression (LSR) in Microsoft Excel to the multiscale cumulative distributions. Manual fitting has been performed as well (results are reported and compared to LSR fitting in Fig. Supplement S1).

#### 3.2.2. Fitting of single-scale cumulative length distributions

Single-scale cumulative length distributions have been analyzed by means of the Maximum Likelihood Estimation (MLE) and Kolmogorov-Smirnov (KS) statistical tests to retrieve the best fitting mathematical function (Dichiarante et al., 2020; Kolyukhin and Torabi, 2013; Rizzo et al., 2017). The mathematical functions considered were negative exponential, power-law and log-normal (Fig. 2b). The advantage of adopting MLE-KS statistical tests derives from the possibility to also retrieve the function parameters (namely the exponent $\lambda$ for the exponential, the exponent $\alpha$ for the power-law and the mean $\mu$ and standard deviation $\sigma$ for the log-normal functions) in addition to the mathematical function best approximating the observed cumulative length distributions. A dedicated MATLAB script implementing the freely available functions provided in the latest version of FracPaQ (Healy et al., 2017; Rizzo et al., 2017) was used to this purpose. The results of the MLE-KS tests are reported in "checkerboard" diagrams, following the method proposed by Dichiarante et al. (2020) (Fig. 2d). Such diagrams allow to image the results of the MLE-KS tests on the selected portions of the cumulative distribution, i.e., the best fitting mathematical function for varying subdomains of the cumulative distribution. A subdomain is defined as a segment of the cumulative distribution curve bounded by a lower and upper cut value (Fig. 2c-d). The upper cut (*UC*) value represents the distance, expressed in terms of percentage of the total number of elements contained in the cumulative distribution, from the shortest observed length. The lower cut (*LC*) value represents the distance, in terms of percentage of the total number of elements contained in the cumulative distribution, from the longest observed length. On the checkerboard diagrams, the *LC*

values are plotted versus the *UC* values (Fig. 2d). Each point of the checkboard represents a specific percentage range of the total cumulative distribution between the upper and lower cut limits over which the MLE-KS tests were run. The plotted symbol represents the mathematical function among those considered (power-law, exponential, log-normal) for which the MLE-KS tests yielded the highest fitting score, whereas the symbol is color-coded according to the retrieved value of the fitting scores (namely the H-percentage *HP* and the P-percentage *PP* parameters, see Rizzo et al., 2017; Dichiarante et al., 2020). This analytical approach allows for the determination of the mathematical function that best fits the truncated cumulative distribution and for the evaluation of the effects of truncation and censoring biases (Fig. 2c-d). We report in the main text and tables the range of *UC* values for which each function fits best: the upper cut values quantify the “truncation” of the cumulative distribution at short lengths, and this has been demonstrated to deeply affect the results of MLE-KS tests (Dichiarante et al., 2020).

### 3.3. Spatial distribution analysis

The spatial distribution of lineaments has been quantified following the approach by Sanderson and Peacock (2019). We analyzed the spacing between lineaments collected along virtual scanlines computed with the NetworkGT toolkit in QGis 3.12.2 (Nyberg et al., 2018) (e.g., Fig. 2e). The lineaments were classified and grouped into orientation sets. A grid of equally spaced virtual scanlines (100 m spacing for the 1:5,000 scale and 500 m spacing for the 1:25,000 and 1:100,000 scales) oriented perpendicular to the selected lineament set orientation was drawn upon the imported lineament map with NetworkGT (e.g., Fig. 2e). Intersections between each virtual scanline and map lineaments were also obtained in NetworkGT. For each scanline, we analyzed the statistics (mean – $\mu$, standard deviation – $\sigma_S$ and minimum and maximum values) for several parameters (Fig. 2e): (i) spacing (*S*) between lineaments; (ii) Coefficient of Variation (*CoV*) of the spacing, defined as the ratio between the standard deviation of spacing along a scanline and its average ($CoV = \sigma_S/\mu$) (Gillespie et al., 2001); (iii) coefficient of heterogeneity ($V_f$) and its statistical significance ($V^*$) according to Sanderson and Peacock (2019). The *CoV* of spacing is commonly adopted to assess the spatial organization (clustering vs. uniform distribution) of lineaments along scanlines (e.g. Gillespie et al., 2001). *CoV* values > 1 are usually related to clustered lineaments; *CoV* = 1 should represent a (negative) exponential-random distribution of spacing intervals, and *CoV* < 1 is usually related to log-normal (uniform) spacing distributions (Gillespie et al., 2001; McCaffrey et al., 2020; Odling et al., 1999). The spacing heterogeneity, i.e., the deviation of the spacing distribution along a scanline from a uniform distribution, is quantified by the $V_f$ and $V^*$ coefficients computed with the Kuiper method (Sanderson and Peacock, 2019). The coefficient of heterogeneity $V_f$ quantifies the deviation from a theoretical uniform distribution of the observed spacing distribution along a given scanline expressed as the sum of the moduli of the positive and negative deviations (see Sanderson and Peacock, 2019; Fig. 2e). The coefficient $V^*$ quantifies the statistical significance of the heterogeneity factor $V_f$:

$$V^* = V_f\left(\sqrt{N_i} + 0.155 + \frac{0.24}{\sqrt{N_i}}\right)$$

where $N_i$ represents the number of lineaments intersected by the scanline. Stephens (1970) demonstrated that for $V^* > 1.75$, 2.0 and 2.3, the null hypothesis of uniformity can be rejected at the 95%, 99% and 99.9% levels, respectively. Thus, the coefficient $V^*$ can be used to quantify the probability that a certain spacing distribution is uniform or not.

We present the results of this analyses as $CoV$-$V^*$ diagrams (Fig. 2f) in which we plot the statistical distribution of the values of $CoV$ vs. $V^*$ as box-and-whiskers. In doing so, we can qualitatively evaluate if, statistically, a set of lineaments has a random or organized spatial distribution (Sanderson and Peacock, 2019). With this method, four main spatial organization types can be distinguished (Fig. 2f): (i) uniform distribution, characterized by $CoV \ll 1$ and $V^* < 1.75$; (ii) random distribution, characterized by $CoV \approx 1$, $V^* < 1.75$; (iii) corridor/clustered distribution, characterized by $CoV > 1$ and $V^* >$ 1.75-2.00; (iv) fractal distribution, characterized by $CoV \gg 1$ and $V^* \gg$ 1.75-2.00. Scanlines with more than 10, 5 and 3 lineament intersections were considered on maps at 1:5,000, 1:25,000, and 1:100,000, respectively.

Given the limited number of intersections recorded by each virtual scanline in our maps ($N_i \ll 30$), more advanced and up-to-date analyses of the spacing variability (e.g., Marrett et al., 2018; Sanderson and Peacock, 2019; Bistacchi et al., 2020) were not possible. The $Vf$-$V^*$ method proposed by Sanderson and Peacock (2019) yields statistically meaningful results for datasets containing a number of intersected lineaments $N_i > 6$. Our datasets unfortunately do not always satisfy this requirement. To perform the analysis on a statistically meaningful number of samples (total number of scanlines, $N_{SL} > 10$), we had to reduce the minimum number of spacing data each scanline had to contain to be included in the dataset. Therefore, our analyses report the results from scanlines showing a minimum number of intersections $N_i$ >10, >5, and >3 for the analyses performed at the 1:5,000, 1:25,000 and 1:100,000 scale, respectively.

All parameters and the related abbreviations are reported in Table 1.

## 4. Results

### 4.1. Lineament maps description

The manual picking of topographic lineaments on different digital representations of the selected areas of Bømlo led to the production of maps at different scales (Fig. 3, Supplement S2-S3). The orthophotos retrieved from UAV-drone surveys and the related lineament map (Fig. 3a, Supplement S2a) helped to characterize the main outcrop of the GFZ along the eastern shoreline of Goddo Island (Ceccato et al., 2021a, 2021b; Viola et al., 2016). The investigated areas extend for 2127 $m^2$ over which we picked 930 lineaments. Lineament mapping on LiDAR DTM and aerial imagery at the 1:5,000 scale (Fig. 3b, Supplement S2b) was performed on the best exposed areas along the coastline of the Goddo Island and nearby smaller islands. The resulting lineament map covers more than 17 $km^2$ and includes 3,835 lineaments. Furthermore, we generated additional lineament maps from the interpretation of the LiDAR DTM at the 1:25,000 and 1:100,000 scales over the same area (83 $km^2$; Fig. 3c-d, Supplement S2c-d). The 1:25,000 lineament map contains 894 lineaments, whereas the 1:100,000 map contains 249 lineaments.

### 4.2. Fractal dimension

The fractal dimension of the lineament maps at all scales was evaluated by applying the box-counting method (Bonnet et al., 2001; Gillespie et al., 1993) (Fig. 3). The number of filled boxes $N_b$ decreases with increasing box size $b$ following a power-law relationship (Fig. 4). The power-law exponents (the fractal exponents) retrieved from the box counting analyses of the lineament maps at different scales ranges between 1.45 and 1.61 (Fig. 4). On average, the lineament network is characterized by a fractal dimension $D = 1.51 \pm 0.14$ ($2\sigma_S$).

### 4.3. Lineament Orientation

The comparison of the rose diagrams at different scales of observation allows to define some dominant trends (Fig. 5a-b). The five main orientation sets are (Fig. 5a, Supplement S3, Table 2): (a) a N-S-striking Set 1; (b) a NE-SW-striking Set 2; (c) a ENE-WSW-striking Set 3; (d) a ESE-WNW-striking Set 4, and (e) a SE-NW-striking Set 5. These sets display a significant variation of their relative abundance across scales. At the smallest scale of observation (1:100,000), Set 5 is dominant, whereas at the largest observation scale (1:100), Sets 1 and 2 are dominant (Fig. 5b). At intermediate scales (1:5,000; 1:25,000), all sets are equally represented (Table 2). Set 3 is the least represented, occurring only in small percentages (<10%) at all scales (Table 2). Sets 2 to 5 have a constant average orientation across scales but the average orientation of Set 1 lineaments changes with scale of observation. N-S-striking orientations are dominant at the smallest and largest scales of observation. At the intermediate scale, Set 1 presents either a NNW- (scale 1:5,000) or a NNE-dominant strike (scale 1:25,000) (Fig. 5a). Therefore, we have subdivided Set 1 into Set 1a, including NNE-SSW-striking lineaments, and Set 1b, including N-S- to NNW-SSE-striking lineaments. This subdivision will be adopted for discussing the spatial organization of the lineaments.

### 4.4. Cumulative length distributions

The results of LSR fitting and MLE-KS tests are reported and summarized in Fig. 6 and Table 3; the checkerboards diagrams are reported in the supplementary material (Fig. Supplement S4).

The results of the MLE-KS tests suggest that a log-normal function best approximates the entire single-scale distribution in all considered cases (Fig. Supplement S4; Table 3; Dichiarante et al., 2020). Variably truncated distributions are best approximated by either negative exponential or power-law functions (Table 3). In particular, the truncated length probability distributions for both single sets and the entire lineament network mapped at 1:100 are best represented by negative exponential functions, with $\lambda$ ranging between 0.65 and 1.25. Truncated distributions retrieved from lineament maps at 1:5,000 are best fitted, in most cases, by power-law functions with a minimum exponent $\alpha$ of 2.2. Truncated length distributions for lineaments mapped at 1:25,000 and 1:100,000 scale are well approximated by negative exponential functions, with an average $\lambda$ of 0.004 and 0.0017, respectively (Table 3).

Figure 6a reports the cumulative length distributions for the entire set of lineament maps normalized to the area of investigation at each scale of observation. The multiscale normalized cumulative distribution obey a general power-law relationship valid over five orders of magnitude (1 m to 10,000 m). The power-law exponent α is 1.88 (Fig. 6a). Figure 6b reports the multiscale cumulative length distributions for each lineament set normalized for the area of investigation. Also in this case, multiscale distributions obey a general power law scaling with a characteristic exponent α for each set ranging between 1.62 and 2.12 (Fig. 6b, Supplement S1).

**4.5. Lineament density and intensity**

As also suggested by the cross-scale variation of the relative proportions of the orientation sets (Fig. 5a-b), also the normalized density $P_{20}$ ($m^{-2}$) and intensity $P_{21}$ ($m/m^2$) of each lineament set vary across scales. The variations of density and intensity are both described by a power-law relationship in log-log diagrams plotting the scale on the X-axis (e.g., $10^5$ = 1:100,000) and the density $P_{20}$ or intensity $P_{21}$ on the Y-axis (Fig. 7a-b) (e.g., Castaing et al., 1996). The variation trend for the total lineament density $P_{20}$ of each map at different scales is characterized by power-law exponents β = 1.77 (Fig. 7a). Sets 1, 2, and 3 display β values larger than the average value; Sets 4 and 5 display β values smaller than the average values. Similarly, the variation trend for $P_{21}$ is characterized by a power-law exponent δ = 0.86 (Fig. 7b); Sets 1, 2, and 3 show δ values larger than the average value. Sets 4 and 5 display δ values smaller than the average value.

**4.6. Spacing and organization at different scales**

The *CoV*-*V** diagrams highlight a similar decreasing trend for all analyzed lineament sets with increasing scale of observation (from 1:100,000 to 1:5,000). At the 1:5,000 scale (Fig. 8a), Set 1a and 1b lineaments are characterized by $CoV \leq 1$ and $V^* \leq 1.75$.; suggesting a random-to-uniform spatial distribution. At smaller scales, *CoV* for Set 1 exhibit a tendency towards random-to-uniform distribution (Fig. 8b-c). Set 2 lineaments display *CoV* on average >1 at the 1:5,000 scale, and $V^*$ >1.75 for a significant number of data (>40% of the total number of data), suggesting a clustered spatial distribution. At smaller scales, both *CoV* and $V^*$ values generally decrease, although some of the analyzed scanlines still display *CoV* >1 and $V^*$ >1.75-2.00. Set 3 lineaments are too scattered and sparce to allow for a meaningful analysis of their spatial arrangement and, therefore, they are not reported in Fig. 8. Set 4 lineaments mapped at the 1:5,000 scale on average show *CoV* values >1, but $V^*$ is rarely >1.75. At smaller scales, both *CoV* and $V^*$ decrease progressively. *CoV* and $V^*$ for Set 5 lineaments are generally <1 and <1.75, respectively. The most significative variation across scales in spatial distribution occurs for Set 2 and Set 4, both of which exhibit a tendency towards clustering at the large scale (1:5,000; $CoV > 1$, $V^* > 1.75$; Figs. 8a-9d), whereas they exhibit a tendency toward a random-to-uniform distribution at the smaller scales (1:25,000 and 1:100,000; $CoV < 1$; $V^* < 1.75$; Figs. 8b-c, 9e). None of the lineament sets show a tendency to develop fractal behavior with a power-law spacing distribution.

## 5. Discussion

In the following, we firstly assess the geometrical characteristics, scaling laws and exponent values obtained for the Rolvsnes granodiorite fracture network. Then, we evaluate the possible biases affecting the analyzed datasets and their effect on the quantification of the fracture network geometrical properties. In addition, we discuss the implications of applying the scaling relationships to the quantification of fracturing and reservoir permeability at different scales by integrating our results with field geological data.

### 5.1. Characterization of geometric properties of the Rolvsnes granodiorite lineament network

The fractal dimensions $D$ retrieved from the analysis of 2D lineament maps clusters around 1.5 (Fig. 4), similar to what is commonly reported from other case studies on lineament pattern fractal dimensions (Bonnet et al., 2001; Hirata, 1989). Also, the normalized cumulative distribution of lineament lengths effectively defines a single scaling law, which can be best described by a power-law relationship with an exponent $\alpha = 1.88$ (Fig. 6a; Table 5). The general scaling law obtained for the overall lineament network is very similar to that derived from many other case studies of fracture networks affecting both crystalline basements and (meta)sedimentary rocks, with an average power-law exponent close to $\alpha = 2$ (cf. Bertrand et al., 2015; Bonnet et al., 2001; Bossennec et al., 2021; Chabani et al., 2021; Le Garzic et al., 2011; McCaffrey et al., 2020; Odling, 1997; Torabi and Berg, 2011). The power-law scaling relationship defined by the lineament density $P_{20}$ values is characterized by a power-law exponent $\beta = 1.77$, similar to the value of 1.7 commonly observed in many other fault networks (Castaing et al., 1996; Bonnet et al., 2001, and references therein).

#### 5.1.1. Scale-invariant lineament network

A similar fractal dimension and power-law scaling relationship are commonly used as evidence for the occurrence of a fracture network whose geometrical properties (size of fractures, i.e., length, and spatial correlation and organization) are scale-invariant (Bonnet et al., 2001). This suggests that, at a first approximation, the documented lineament pattern in the Rolvsnes granodiorite is self-similar at any scale of observation. However, the Rolvsnes granodiorite case appears to be more complex than it would seem at a first glance. The Rolvsnes granodiorite lineament network is composed of five main orientation sets with variable relative abundance, density, and intensity across scales (Figs. 5 and 7, Table 2). The observed variations of density and intensity are predictable and can be described by a general power-law scaling, the exponent of which is characteristic of each orientation set (Fig. 7; Table 5). Even though the single-scale cumulative length distribution for each orientation set can be best approximated by other scaling laws than power-law (Table 3), the multiscale cumulative length distribution is best approximated by a power-law scaling relationship (Fig. 5b; Table 5). Again, each orientation set is characterized by its own power-law exponent (Fig. 5b; Table 5), which differs slightly from that computed for the entire lineament network.

#### 5.1.2. Lineament Types within a hierarchical fracture network

Some lineament sets display similar trends of variation of the relative abundance and intensity, such that they can be grouped into two main set types (Figs. 6b, 9; Table 5): (1) Type A includes Sets 1, 2 and 3, characterized by comparable $P_{20}$ and $P_{21}$ variation trends across scales ($\beta \approx 1.90$; $\delta \approx 0.95$) and length distributions ($\alpha \geq 2$); (2) Type B includes Sets 4 and 5, characterized by comparable $P_{20}$ and $P_{21}$ variation trends across scales ($\beta \approx 1.60$; $\delta \approx 0.70$) and length distributions ($\alpha < 1.7$). This classification into Type A and B lineament sets is not directly reflected in the *CoV-V** diagrams (Fig. 8a-b). they rather suggest a scale-dependent organization of spacing for each lineament set.

Therefore, the observed density/intensity variation and spatial organization trends indicate the occurrence of a hierarchical (scale-dependent) organization of lineament sets within a network presenting overall scale-invariant geometrical properties (e.g., Le Garzic et al., 2011). In this hierarchy, Type B lineaments represent the higher-order structures, controlling the geometrical properties of the network at the regional scale (Fig. 9d-e). Type A lineaments represent lower-order structures and control the geometrical properties of the network at the local-to-outcrop scale (Fig. 9d-e).

At the smallest investigated scale, the homogeneously spaced, WNW-to-NW-striking Type B lineaments (Fig. 9d-e) dominate the network. These lineaments are characteristic of – and predominant over – the whole of onshore western and southwestern Norway (Gabrielsen et al., 2002; Gabrielsen and Braathen, 2014; Tartaglia et al., 2022), as well as offshore (Preiss and Adam, 2021). The power-law exponent for Type B lineaments ($\alpha < 1.7$) suggests that long lineaments represent a substantial part of the overall lineament population. These observations also suggest that Type B lineaments probably result from the homogeneous distribution of deformation at the regional-scale, while still representing localized zones accommodating significant deformation at the outcrop-scale, when compared to Type A structures (Ackermann et al., 2001). Therefore, these lineaments probably represent major fractures and normal fault zones formed and repeatedly reactivated during the prolonged brittle tectonic history of the Rolvsnes granodiorite (Ceccato et al., 2021a; Preiss and Adam, 2021; Scheiber and Viola, 2018; Viola et al., 2016). The schematic representation of lineaments in Fig. 9d-e highlights an heterogeneous distribution of Type B lineaments, which is not captured by the statistical analysis of spacing heterogeneity. Indeed, the Rolvsnes granodiorite can be subdivided into several domains of the lineament maps where either Set 4 or Set 5 lineaments are predominant at the regional scale ("Set 4-5 domain" – grey and dashed transparent areas in Fig. 9d-e). These domain-type distribution of regional lineaments were already reported by Scheiber and Viola (2018). At the largest analyzed scale, the lineament network is mainly dominated by random-to-clustered, NNW-SSE to NE-SW-striking Type A lineament sets (Fig. 9d-e). Accordingly, the general power-law exponent ($\alpha \geq 2$) suggests that, among Type A lineaments, short lineaments represent a significant part of that population, probably resulting from an incipient stage of distributed faulting and deformation accommodation (e.g., Ackermann et al., 2001).

### 5.2. Analysis of reliability, biases ad limitations of the scaling laws

The scaling laws described here have several limitations in their applicability related to: (i) occurrence of different orientation sets; (ii) network heterogeneity at different scales; (iii) analytical biases.

#### 5.2.1. Reliability and biases behind orientation set definitions

The lineament sets defined by this study are grouped in classes based on their azimuthal orientation. As such, they may or may not share a genetic relationship. However, field analyses (Scheiber et al., 2015, 2016; Scheiber and Viola, 2018) have demonstrated the occurrence of systematic sets of fractures, genetically related in terms of chronology, tectonic phase, and orientation, which can be identified from remote sensing techniques. Thus, we assume that the identified orientation sets effectively represent groups of genetically correlated fractures (see below, Section 5.3).

The lineament orientation reported by the 1:100 rose diagram differs significantly from the orientations of all other diagrams (Fig. 5). Even though the number of lineaments interpreted from UAV-drone imagery is statistically significant ($N_{Lin}$ = 930), the N-S- trending outcrop exposure, its 3D topography, and the location of the exposed area along a major fault zone (Ceccato et al., 2021a, 2021b) are such that it is necessary to question whether the obtained results are truly representative of the larger-scale lineament network. The observed variations of density, intensity, and relative abundance of orientation sets across scales could be affected by several analytical and interpretative orientation-dependent and classification biases.

First of all, under-sampling of specific lineament orientations during manual interpretation may be due to (Scheiber et al., 2015): (i) interpretative biases of the operator; (ii) changes in resolution of the digital representation of the terrain (DTMs and orthophotos) with the changing scale of observation; (iii) constant direction of the light source adopted for the LiDAR DTM hill shading (from NW in our study). The change in resolution would affect equally each orientation set, thus maintaining a constant relative abundance across scales. Likely, constant direction of the light source may affect the detection of lineaments at specific orientations, but systematic effects have not been identified by previous studies (Scheiber et al., 2015). Also, the topography and exposure of outcrops with rough topography (such as the GFZ outcrop analyzed here at 1:100 scale) might affect the exposure and detection of specific lineaments and thus orientation sets – some sets may be more visible than others. In particular, gently dipping fractures or fractures parallel to the surface of the outcrop might be underrepresented. Moreover, rose diagrams only report the number of lineaments, and do not consider their spatial persistence (length), such that fractures related to the main GFZ and expected to be dominant at the local scale might be represented by very few, but longer lineaments. Thus, the small number of lineaments can be diluted and obscured by the large number of short lineaments related to background fracturing.

This notwithstanding, by considering the relative abundance of orientation sets retrieved only from the 1:5,000; 1:25,000 and 1:100,000 lineament maps, we can still observe the relative decrease of Sets 1, 2, 3 along with the increase of Set 5 lineaments at progressively smaller scales of observation (Fig. 4b). This suggests that the observed variation trends reflect an

effective variation in relative abundance of lineaments across scales and represent a real characteristic of the lineament network.

#### 5.2.2. Network vs. orientation set scaling laws

The studied lineament network exhibits some general power-law relationships describing the multiscale behavior of both lineament length distribution ($\alpha$ = 1.88), density $P_{20}$ ($\beta$ = 1.77) and intensity $P_{21}$ ($\delta$ = 0.86) (Table 5). These general power-law scaling laws may effectively be adopted to retrieve fracture network properties (geometrical properties and permeability) at any scale of observation. However, the adoption of a general scaling law for the geometrical properties without taking into consideration the peculiarity of each orientation set building up the network, may lead to an erroneous extrapolation of the analyzed properties. For example, lineament sets exhibit different power-law exponents for density $P_{20}$ and intensity $P_{21}$ distributions, which, in our case study are systematically smaller for Type A sets, and larger for Type B sets than the exponent of the network taken as a whole (Fig. 7). Adopting power-law exponents larger than the actual exponent of the individual lineament set would lead to an overestimation of the network properties at larger scales, and vice-versa. In the case of the Rolvsnes granodiorite lineament network, this overestimation/underestimation could be significant and reach one order of magnitude in terms of intensity and density (Fig. 7).

#### 5.2.3. Spatial organization and scaling limitations

Field investigations (Ceccato et al., 2021a, 2021b; Scheiber and Viola, 2018) have revealed the highly heterogeneous distribution of fractures at the outcrop scale. Most of the identified fracture sets at the outcrop occur with either a clustered spatial organization or a variable intensity over short distances (50-100 m; Ceccato et al., 2021a). This clearly represents a limitation to the acritical extrapolation of the general power-law determined in this study, and thus the lower bound for the application of the proposed power-law scaling (Bonnet et al., 2001). Similarly, the “zonal” spatial distribution of Set 4 and 5 lineaments identified at small scale of observation (see the identified “Set 4 domain” and “Set 5 domain” reported in Fig. 9d-e) needs to be accounted for when evaluating the upper limit of applicability of the general scaling laws defined here. The outcrop-scale spatial heterogeneity and the overestimation/underestimation effects due to applying a general power-law scaling become relevant when considering the role that different fracture sets may have in the definition of the net permeability of a fractured crystalline basement, as highlighted by field studies (Ceccato et al., 2021a, 2021b; Gabrielsen and Braathen, 2014; Torabi et al., 2018).

#### 5.2.4. Analytical biases and statistical methods to overcome them

In the light of the discussion above, it is therefore important to analyze the mapped distributions with appropriate methods and to always consider the regional lineament network as composed of different orientation sets, each of which is characterized by its own geometrical and scaling properties. Additionally, we need to consider all potential sources of error and estimate the analytical/human biases that potentially affect our analyses.

It is worth noting that, manual fitting of our multiscale distribution slightly overestimates the power-law exponents retrieved from LSR methods (Fig. Supplement S1). This might be since LSR methods consider the entire distribution, including the portions of the distributions affected by censoring and truncation, which are inherently excluded by manual fitting.

MLE-KS tests have already demonstrated their strength in the analysis of fault attribute distributions (Dichiarante et al., 2020; Kolyukhin and Torabi, 2013). In our case, the results of length cumulative distributions fitting with MLE-KS tests differ significantly from the multiscale LSR power-law relations reported in Fig. 6. Even when the distribution is best approximated by a power-law function (e.g., most lineament sets mapped at 1:5,000; Table 3), the values of the power-law exponents retrieved from MLE-KS tests ($\alpha$ >2.2) differ from those obtained from the LSR fitting of multiscale distributions ($\alpha$ < 2.12).

These deviations (both that of single-scale distributions from the power-law, and that of the single-scale power-law exponents from the multiscale ones) are commonly observed in almost all natural fracture networks. Remarkable deviations from a power-law scaling behavior have been previously explained as resulting from several causes: (i) analytical biases (such as truncation and censoring of lineaments; Dichiarante et al., 2020; Manzocchi et al., 2009; Odling, 1997; Yielding et al., 1996); (ii) subdivision of long lineaments into segments (segmentation; Ackermann et al., 2001; Cao and Lei, 2018; Scholz, 2002; Schultz et al., 2013; Xu et al., 2006); (iii) effectively different scaling properties at different scales of observation (Castaing et al., 1996; Le Garzic et al., 2011; Kruhl, 2013).

Power-law fitting is usually retrieved from short segments in the central portions of a “truncated” cumulative distribution (e.g. Dichiarante et al., 2020). Truncation and censoring biases may affect large portions (even >50% of data) of the cumulative distribution. This would mean that most (>>50% of data) of the analyzed dataset is biased, and thus of little use to any kind of statistically significant analysis, such that the related fitting results are not statistically meaningful. The multiscale distribution analysis can reveal the overarching scaling law of the fracture network and of each lineament set, but the actual values of the exponents of the fitting laws need to be carefully evaluated by also considering the statistical significance of the analyzed dataset.

Segmentation of long lineaments into shorter segments may be due to several causes, both introduced into the dataset by analytical/interpretative biases, and intrinsically related to the network topology, fracture chronology and geological fracture formation processes. Segmentation may result from partial exposure and cover of the fracture network, and it may increase the power-law scaling exponent, without affecting the type of scaling-law function (Cao and Lei, 2018). Segmentation may be related to the progressive growth stages of fault/joint patterns evolving with increasing accommodated deformation and faulting maturity from a network composed of completely isolated short fractures to a network formed by a few long, single lineaments, through fracture interaction and interconnection (Ackermann et al., 2001; Michas et al., 2015; Scholz, 2002). This has been demonstrated to affect both the shape of the mathematical function describing the length distribution (exponential vs. power-law), as well as the power-law exponent at a specific scale of observation (Schultz et al., 2013). However, this may explain the difference in scaling relationships observed during the evolution of a fracture network through time but not at different scales of observation. In addition, the subjective choice of tracing single segments

composing a longer lineament as separate fractures rather than tracing a single, continuous, long lineament, may likely affect the cumulative length distributions of the fracture network. Tracing single segments would increase the number of short segments compared to longer segments, at constant $P_{21}$ intensity, increasing the total number of traced lineaments, and thus increasing the power-law exponent of the distribution (Xu et al., 2006). This segmentation bias may justify the fact that power-law exponents of the LSR multiscale length distributions of each fracture set (Fig. 5b) are systematically smaller than those obtained from single-scale MLE-KS tests at 1:5,000 scale. Whether or not this sampling bias may effectively affect the mathematical shape of the cumulative distribution would deserve further investigations, which goes beyond the scope of the present paper.

Thus, the most plausible option is that the fracture network may effectively present different scaling properties at different scales of observations (Kruhl, 2013). Indeed, fault and fracture networks may exhibit a hierarchical organization, which inherently implies scale-dependent geometrical properties and spatial distribution of lineaments (Castaing et al., 1996; Le Garzic et al., 2011). In fact, this is also consistent with the observed variation of relative abundances of orientation sets across scales: each lineament set contributes differently to the overall fracture network geometrical characters and thus the variation of the relative abundance may also lead to variations in geometrical properties (spatial organization and length distributions) at different scales (e.g., Le Garzic et al., 2011).

### 5.3. Integration of remote sensing and field observations

The fracture pattern of the Rolvsnes granodiorite includes three main classes of fractures and fault zones (Scheiber et al., 2016; Scheiber and Viola, 2018): (i) pre-Permian, ESE-WNW and NE-SW striking mineralized shear fractures and minor faults; (ii) Permo-Jurassic major normal faults, mainly striking NW-SE and N-S; (iii) Cretaceous fracture clusters striking N-S to NNE-SSW.

Smaller normal faults and mineralized shear fractures described by Scheiber and Viola (2018) are subparallel to the lineament Sets 2-3-4 defined here. They form the background fracture pattern of the Rolvsnes granodiorite (Ceccato et al., 2021a). In particular, Set 4 lineaments are subparallel to the ESE-WNW orientation of the (relatively) oldest generation of fractures identified in the field (Scheiber and Viola, 2018). Indeed, the regional distribution of Set 4 and the geometrical characteristics discussed above stress their importance as regional structures accommodating significant deformation through the brittle deformation history of the Rolvsnes granodiorite (Scheiber and Viola, 2018). Conversely, NE-SW minor faults and shear fractures, subparallel to Sets 2 and 3 lineaments, accommodated only limited deformation, which may be compatible with their local-scale distribution and geometrical characteristics (e.g., Ackermann et al., 2001). Permo-Jurassic normal fault zones are oriented NW-SE to N-S, similarly to our Set 5 and Set 1(b) lineaments. The geometrical characters and spatial distribution of Set 5 lineaments suggest their role as important zones of deformation accommodation at the regional scale. N-S to NNE-SSW fracture clusters are comparable to Set 1(a) orientations; they are indeed local-scale structural features and are inferred to have accommodated limited deformation during Cretaceous rifting of the North Sea (Scheiber and Viola, 2018).

Summarizing, although our lineament classification in Type A and B is certainly an oversimplification of the actual complexity of the natural fracture network, it provides valuable information as to the geometrical characteristics of faults and fractures and their regional/local importance as zones of deformation accommodation.

### 5.3.1. Constraints on the multiscale permeability structure of crystalline basement

As constrained by field studies, each lineament/fracture set may contribute differently to the bulk permeability of a fractured crystalline basement block (Ceccato et al., 2021a, 2021b; Gabrielsen and Braathen, 2014; Torabi et al., 2018). The effects of the background fracturing of the Rolvsnes granodiorite on permeability is secondary (Ceccato et al., 2021a), it being mainly composed of minor sealed faults and mineralized fractures belonging to Set 2-3-4 lineaments. It is the regional scale structures, like our Type B lineaments (Set 5; e.g., the GFZ) that effectively control permeability, fluid flow, and reservoir compartmentalization at the regional scale (Ceccato et al., 2021a; Holdsworth et al., 2019). Results from in-situ petrophysical analyses and discrete fracture network modelling of fault zone permeability have shown that these structures behave as mixed conduit-barrier for fluid flow, and are characterized by a strongly anisotropic permeability tensor (Caine et al., 1996; Ceccato et al., 2021a). Fluid flow is promoted parallel to the main fault plane, especially parallel to the (sub-horizontal) intersection directions of the dominant fracture sets within the fault damage zone, whereas the anisotropic permeability of the fault core brittle structural facies buffers cross-fault fluid flow (Ceccato et al., 2021a, 2021b; Tartaglia et al., 2020). Conversely, fracture clusters, comparable to our Set 1(a) lineaments, may represent effective fluid pathways at the outcrop scale, acting as preferential conduits for vertical fluid flow within the basement (Ceccato et al., 2021a, 2021b; Place et al., 2016; Souque et al., 2019; Torabi et al., 2018).

In summary, by integrating field and remote sensing data we can improve the conceptual models and their dimensioning in an attempt to describe the anisotropic permeability structure of a fractured crystalline basement at different scales (e.g., Fig. 11 of Ceccato et al., 2021b). Our results constrain the heterogeneous structure of a fractured basement block in terms of orientation and spatial distribution of permeability. The permeability of the fractured basement at the regional-scale is characterized by the occurrence of rhombohedral-shaped compartments (the fault-bounded polygonal domains of Ceccato et al., 2021b) that are homogeneously distributed and defined by the higher-hierarchical order Type B lineaments. Their extension is determined by the spacing of Type B lineaments, ranging in the order of 500-1,000 m at the regional scale (Table 4). Fluid flow is promoted along the major fault planes and parallel to the sub-horizontal intersections of fracture sets dominant within the fault damage zones (Ceccato et al., 2021a). Within these rhombohedral compartments, permeability is heterogeneously distributed at the 50-100 m scale, following the random-to-uniform spacing distribution of lower-hierarchical order Type A (Set 1b) lineaments (Table 4). At the outcrop scale, these N-S lineaments are represented by fracture clusters, which promote vertical fluid flow (Ceccato et al., 2021a, 2021b).

Accordingly, any underestimation/overestimation of spatial distribution and density of the lineaments may deeply affect the accuracy of hydrological and petrophysical models of fractured basement blocks at the outcrop and at the sub-seismic resolution scale (Bertrand et al., 2015; Le Garzic et al., 2011).

## 6 Conclusions

The fractured crystalline basement of the Rolvsnes granodiorite on Bømlo is characterized by the occurrence of a fractal fracture network controlled by a general power-law scaling law for the distribution of fracture lengths. However, detailed orientation-dependent analyses have revealed that this first-approximation scale-invariant lineament network is composed of lineament sets, which individually exhibit a scale-dependent hierarchical spatial distribution, and parameter variation trends with the scale of observation. Different trends of intensity/density variation across scales for each orientation set have been detected, as well as different scaling laws for length distribution of each orientation set. These results, integrated with field observations, suggest that the documented lineament network results from the summation of different geological structures (e.g., faults vs. joints, major fault zones vs. incipient minor faults) organized in a hierarchical manner and characterized by different geometrical and scale-dependent properties.

The hierarchical lineament network affecting the Rolvsnes granodiorite controls the anisotropy and directionality of the permeability structure of the basement at different scales. At the regional scale, the crystalline basement is characterized by a rhombohedral pattern of basement compartments bounded by regional fault zones impermeable to cross-fault fluid flow. Within these compartments, the permeability structure is controlled by local-scale fracture clusters, promoting subvertical N-S striking fluid flow.

Our study allows us to draw some general conclusions about the methods for characterization of fracture network and their analysis:

- Firstly, the presented multiscale analytical workflow may represent a valid option for the quantification of large, inherently incomplete (due to analytical and subjective biases) lineament datasets. The lineament maps retrieved from digital terrain and surface models of the Rolvsnes granodiorite offer very large datasets, which are inherently incomplete due to partial exposure and/or incomplete sampling of lineament due to resolution or human biases. Thus, a statistical approach such as that proposed in this paper is highly recommended when aiming to retrieve relevant information from datasets that, for several reasons, are only partially representative for the entire fracture network.
- Detailed orientation-dependent, multiscale analyses of the lineament network can provide the different scaling laws and geometrical properties for each constituent lineament set, which can be adopted to improve the detail and tune the accuracy of permeability models of fractured crystalline basements considering outcrop-scale structural features.

- The integration of multiscale length distribution analyses, multiscale intensity/density estimations and multiscale description of spatial organization provides useful information for the classification of topographic lineaments as different geological structures (e.g., fracture clusters vs fault zones) with specific hierarchy and control on the permeability of the fractured basement.

**Author Contribution**

A.C.: Conceptualization, Data curation, Formal analysis, Investigation, Methodology, Validation, Visualization, Writing – Original draft preparation. G.T.: Data Curation, Validation, Visualization, Writing – Review and Editing; M.A.: Conceptualization, Data Curation, Validation, Visualization, Writing – Review and Editing; G.V.: Conceptualization, Data Curation, Validation, Visualization, Writing – Review and Editing; Funding acquisition, Project administration.

**Data Availability**

Data analyzed (shapefiles of manually picked lineaments and related geometrical properties) in the present paper are available at: ("Multiscale_lineament_analyses_dataset", Mendeley Data, V1, https://data.mendeley.com/datasets/4zdjpmr9jk/1 - doi: 10.17632/4zdjpmr9jk.1).

**Competing Interests**

The authors declare that they have no conflict of interest.

**Acknowledgments**

Our research work was funded by the still ongoing BASE 2 project ("Basement fracturing and weathering onshore and offshore Norway—Genesis, age, and landscape development" – Part 2) and BASE 3 (NFR grant number 319849), a research initiative launched and steered by the Geological Survey of Norway and supported by Equinor ASA, Aker BP ASA, Lundin Energy Norway AS, Spirit Energy Norway AS, Wintershall Dea Norge, and NGU. We thank all BASE colleagues for continuous discussion and constructive inputs. Roberto Emanuele Rizzo is warmly thanked for fruitful discussions. Eric James Ryan is thanked for field support and the acquisition of UAV-drone imagery. Roy Gabrielsen and an anonymous reviewer are thanked for constructive inputs to an earlier version of the manuscript.

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

## Tables

| Parameter | | Unit |
|---|---|---|
| $\mu$ | Mean | - |
| $\sigma_s$ | Standard deviation | - |
| $R^2$ | Coefficient of determination | - |
| $m$ | Proportionality coefficient for scaling laws | - |
| $s$ | Scale | - |
| $b$ | Box size for box-counting | m |
| $N_{Lin}$ | Number of lineaments | - |
| $N_b$ | Number of filled boxes | - |
| $N_{(l>L)}$ | Number of lineaments with a length l > L | - |
| $N_i$ | Number of lineaments intersected by a scanline | - |
| $N_{SL}$ | Number of scanlines | - |
| $L$ | Length of lineaments | m |
| $D$ | Fractal Dimension | - |
| $P_{20}$ | Lineament-Fracture density | $m^{-2}$ |
| $P_{21}$ | Lineament-Fracture intensity | $m/m^2$ |
| $S$ | Spacing | m |
| $\alpha$ | Exponent of power-law distribution | - |
| $\beta$ | Exponent of density scaling law | - |
| $\delta$ | Exponent of intensity scaling law | - |
| $\lambda$ | Exponent of negative exponential distribution | $m^{-1}$ |
| *Xmin* | Minimum lineament length of the truncated dataset | m |
| *UC* | Upper cut | % |
| *LC* | Lower cut | % |
| *HP* | H-percentage | % |
| *PP* | P-percentage | % |
| *CoV* | Coefficient of Variation for spacing | - |
| *d* | Difference from uniform distribution | % |
| $V_f$ | Coefficient of heterogeneity | - |
| $V^*$ | Statistical significance of coefficient $V_f$ | - |

**Table 1. Summary table of the parameters and related nomenclature adopted in this paper.**

| | Set | Azimuth (°) | $N_{Lin}$ | Relative frequency (%) | $P_{20}$ (m$^{-2}$) | Total $L$ (m) | $P_{21}$ (m/m$^2$) |
|---|---|---|---|---|---|---|---|
| ***s*** | **1:100** | | | | | | |
| **Area (m$^2$)** | **1** | 0-17; 156-180 | 274 | 29.46 | 1.29E-01 | 490.10 | 2.30E-01 |
| 2127 | **2** | 19-67 | 374 | 40.22 | 1.76E-01 | 771.86 | 3.63E-01 |
| | **3** | 68-91 | 97 | 10.43 | 4.56E-02 | 156.25 | 7.35E-02 |
| | **4** | 92-120 | 86 | 9.25 | 4.04E-02 | 134.12 | 6.31E-02 |
| | **5** | 121-155 | 99 | 10.65 | 4.65E-02 | 175.30 | 8.24E-02 |
| | **Total** | | 930 | | 4.37E-01 | 1727.63 | 8.12E-01 |
| | | | | | | | |
| ***s*** | **1:5,000** | | | | | | |
| **Area (m$^2$)** | **1** | 0-28; 154-180 | 1059 | 27.61 | 6.17E-05 | 103655.02 | 6.04E-03 |
| 17170533 | **2** | 29-60 | 896 | 23.36 | 5.22E-05 | 76172.72 | 4.44E-03 |
| | **3** | 61-88 | 299 | 7.80 | 1.74E-05 | 23934.34 | 1.39E-03 |
| | **4** | 89-130 | 832 | 21.69 | 4.85E-05 | 80080.14 | 4.66E-03 |
| | **5** | 131-153 | 749 | 19.53 | 4.36E-05 | 90759.34 | 5.29E-03 |
| | **Total** | | 3835 | | 2.23E-04 | 374601.56 | 2.18E-02 |
| | | | | | | | |
| ***s*** | **1:25,000** | | | | | | |
| **Area (m$^2$)** | **1** | 0-25; 162-180 | 216 | 24.16 | 2.60E-06 | 93912.71 | 1.13E-03 |
| 83000724 | **2** | 26-59 | 187 | 20.92 | 2.25E-06 | 60252.45 | 7.26E-04 |
| | **3** | 60-90 | 62 | 6.94 | 7.47E-07 | 20544.88 | 2.48E-04 |
| | **4** | 91-131 | 187 | 20.92 | 2.25E-06 | 74024.49 | 8.92E-04 |
| | **5** | 131-161 | 242 | 27.07 | 2.92E-06 | 113476.26 | 1.37E-03 |
| | **Total** | | 894 | | 1.08E-05 | 362210.79 | 4.36E-03 |
| | | | | | | | |
| ***s*** | **1:100,000** | | | | | | |
| **Area (m$^2$)** | **1** | 0-19; 171-180 | 48 | 19.28 | 5.78E-07 | 35972.83 | 4.33E-04 |
| 83000724 | **2** | 20-60 | 48 | 19.28 | 5.78E-07 | 38877.02 | 4.68E-04 |
| | **3** | 61-90 | 5 | 2.01 | 6.02E-08 | 4354.50 | 5.25E-05 |
| | **4** | 91-130 | 45 | 18.07 | 5.42E-07 | 36742.25 | 4.43E-04 |
| | **5** | 131-171 | 103 | 41.37 | 1.24E-06 | 106590.06 | 1.28E-03 |
| | **Total** | | 249 | | 3.00E-06 | 222536.66 | 2.68E-03 |

**Table 2. Table presenting the orientation data for the identified lineament sets for each scale of observation. Azimuth (°) represents the average strike of the manually picked lineaments. *Total L* represents the sum of the length of all picked lineaments in each map for each set.**

| | | | | | | Exponential | | | | Power-law | | | | Log-Normal | | | |
|---|---|---|---|---|---|---|---|---|---|---|---|---|---|---|---|---|---|
| *s* | Set | $N_{Lin}$ | *L* min (m) | *L* max (m) | Mean *L* (m) | Fitting Score | Range (UC %) | λ | Xmin (m) | Fitting Score | Range (UC %) | α | Xmin (m) | Fitting Score | Range (UC %) | μ | $\sigma_S$ |
| 1:100 | 1 | 274 | 0.3 | 6.9 | 1.8 | >90 | 40-90 | 1.05-1.25 | >1.4 | - | - | - | - | >90 | 0-35 | 0.45-0.75 | 0.52-0.37 |
| | 2 | 374 | 0.3 | 8.7 | 2.1 | >90 | >10 | 0.65-0.79 | >0.9 | - | - | - | - | >90 | 0-5 | 0.55-0.65 | 0.55-0.57 |
| | 3 | 95 | 0.4 | 5.6 | 1.6 | >90 | 10-65 | 0.95-1.05 | 0.8-1.8 | >90 | 70-85 | 3.8-5 | 1.9 | >90 | 0-5 | 0.35-0.45 | 0.5-0.55 |
| | 4 | 86 | 0.3 | 5.0 | 1.6 | >90 | >50 | 1.05-1.25 | >1.25 | - | - | - | - | >90 | 0-45 | 0.25-0.75 | 0.35-0.65 |
| | 5 | 99 | 0.3 | 10.7 | 1.8 | >90 | 15-65 | 0.90-1.05 | 0.9-2.0 | >90 | >70 | 4.25-4.75 | 2.1 | >90 | 0-10 | 0.4-0.5 | 0.46-0.55 |
| | Total | 930 | 0.3 | 10.7 | 1.9 | >90 | 15-60 | 0.78-0.82 | 1.2-2.0 | >90 | >65 | 3.6-4.1 | 2.1 | >90 | 0-10 | 0.45-0.65 | 0.51-0.59 |
| 1:5,000 | 1 | 1059 | 14.4 | 1759.3 | 97.9 | <90 | 15-20 | 0.013 | 40-50 | >90 | >25 | 2.2-3.1 | 50 | <90 | 0-10 | 4.3-4.5 | 0.63-0.72 |
| | 2 | 896 | 4.8 | 1315.6 | 85.0 | <90 | 10-20 | 0.016 | 40 | >90 | >25 | 2.4-3.2 | 50 | <90 | 0-5 | 4.25 | 0.67-0.70 |
| | 3 | 299 | 5.3 | 1265.5 | 80.0 | <90 | 10-20; 40-55 | 0.017; 0.015 | 40; 60 | <90 | 25-35; >60 | 2.6; 2.8-3.4 | 50; >60 | >90 | 0-5 | 4.2 | 0.65-0.7 |
| | 4 | 832 | 8.1 | 1014.3 | 96.3 | <90 | 10-40; 55-65 | 0.013; 0.0115 | 45; 80 | <90 | 45-55; >70 | 2.6; 2.9 | 50; 90 | >90 | 0-5 | 4.3 | 0.7 |
| | 5 | 749 | 11.8 | 2306.4 | 121.2 | <90 | 25 | 0.01 | 60 | <90 | >30 | 2.4-2.8 | 70 | <90 | 0-20 | 4.5-4.8 | 0.65-0.77 |
| | Total | 3791 | 4.8 | 2306.4 | 97.9 | - | - | - | - | <90 | >45 | 2.4-3.2 | 70 | <90 | 0-40 | 4.3-4.7 | 0.55-0.75 |
| 1:25,000 | 1 | 216 | 88.7 | 1681.5 | 434.8 | >90 | >5 | 0.0036-0.0043 | 200 | - | - | - | - | >90 | 0-5 | 5.5-6 | 0.45-0.55 |
| | 2 | 187 | 61.2 | 1264.1 | 322.2 | >90 | >10 | 0.0045-0.0052 | 175 | - | - | - | - | >90 | 0-5 | 5.6-5.7 | 0.54-0.57 |
| | 3 | 62 | 105.0 | 1678.7 | 331.4 | >90 | 0-10; >75 | 0.0045; 0.003 | 125; 275 | >90 | 15-70 | 2.6-2.9 | 150 | - | - | - | - |
| | 4 | 187 | 79.0 | 2652.4 | 395.9 | >90 | 10-70 | 0.0038-0.004 | 200-400 | >90 | >75 | 3.6-3.8 | 425 | >90 | 0-5 | 5.8-5.9 | 0.55-0.57 |
| | 5 | 242 | 89.9 | 1946.3 | 468.9 | >90 | 5-75 | 0.0032-0.0037 | 250 | >90 | >80 | 3.8-4.6 | 600 | >90 | 0 | 6 | 0.55 |
| | Total | 894 | 61.2 | 2652.4 | 405.2 | >90 | >10 | 0.0034-0.004 | 200 | - | - | - | - | >90 | 0-5 | 5.8-6 | 0.55-0.57 |
| 1:100,000 | 1 | 48 | 312.5 | 1892.8 | 749.4 | >90 | >35 | 0.0029-0.0034 | 650 | - | - | - | - | >90 | 0-30 | 6.5-6.7 | 0.34-0.41 |
| | 2 | 48 | 226.2 | 2683.6 | 809.9 | >90 | 0-15; 30-90 | 0.0016-0.0019 | 300 | - | - | - | - | >90 | 20-25 | 6.7 | 0.5 |
| | 3 | 5 | 400.4 | 1586.0 | 870.9 | - | - | - | - | - | - | - | - | - | - | - | - |
| | 4 | 45 | 208.5 | 3936.9 | 816.5 | - | - | - | - | - | - | - | - | - | - | - | - |
| | 5 | 103 | 250.0 | 3224.7 | 1034.9 | >90 | >15 | 0.0016-0.0018 | 300 | - | - | - | - | >90 | 0-10 | 6.8-6.9 | 0.47-0.55 |
| | Total | 249 | 208.5 | 3936.9 | 893.7 | >90 | >10 | 0.0017-0.00185 | 400 | - | - | - | - | >90 | 0-5 | 6.7-6.8 | 0.5-0.55 |

**Table 3. Summary table of MLE-KS test results on distribution fitting. The Table reports only results of the analyses of datasets containing a significant number of lineaments ($N_{Lin} \geq \sim 50$). Fitting score: value of the *HP* parameter obtained from the MLE-KS tests. *Range (UC%)*: range of Upper Cut values within which the fitting score is maximum for the selected fitting function. *Xmin*: minimum length above which the fitting is performed, corresponding to the lower bound of the Range of Upper Cut values.**

| | *s* | **1:5,000** | | | **1:25,000** | | | **1:100,000** | | |
|---|---|---|---|---|---|---|---|---|---|---|
| | **Set** | ***S* (m)** | ***CoV*** | ***V**** | ***S* (m)** | ***CoV*** | ***V**** | ***S* (m)** | ***CoV*** | ***V**** |
| | | $N_{SL}(N_i>10) = 17$ | | | $N_{SL}(N_i>5) = 11$ | | | $N_{SL}(N_i>3) = 61$ | | |
| μ | 1a | 132.64 | 0.91 | 1.45 | 342.29 | 0.87 | 1.77 | 738.25 | 0.42 | 1.40 |
| $\sigma_S$ | | 78.55 | 0.28 | 0.43 | 96.80 | 0.27 | 0.12 | 314.96 | 0.22 | 0.21 |
| **Min** | | 19.37 | 0.53 | 0.92 | 269.18 | 0.57 | 1.45 | 262.68 | 0.03 | 1.03 |
| **Max** | | 232.39 | 1.69 | 2.30 | 568.70 | 1.42 | 1.85 | 1253.90 | 0.88 | 1.80 |
| | | $N_{SL}(N_i>10) = 38$ | | | $N_{SL}(N_i>5) = 53$ | | | - | | |
| μ | 1b | 189.33 | 0.93 | 1.55 | 487.49 | 0.78 | 1.51 | - | - | - |
| $\sigma_S$ | | 52.19 | 0.18 | 0.37 | 175.44 | 0.23 | 0.37 | - | - | - |
| **Min** | | 87.87 | 0.57 | 0.78 | 133.91 | 0.25 | 0.98 | - | - | - |
| **Max** | | 311.72 | 1.44 | 2.38 | 883.45 | 1.25 | 2.26 | - | - | - |
| | | $N_{SL}(N_i>10) = 47$ | | | $N_{SL}(N_i>5) = 47$ | | | $N_{SL}(N_i>3) = 59$ | | |
| μ | 2 | 133.39 | 1.21 | 1.78 | 475.32 | 0.84 | 1.44 | 1013.41 | 0.71 | 1.52 |
| $\sigma_S$ | | 45.48 | 0.32 | 0.44 | 264.12 | 0.20 | 0.27 | 304.75 | 0.28 | 0.27 |
| **Min** | | 37.64 | 0.59 | 1.15 | 89.24 | 0.37 | 0.87 | 352.64 | 0.28 | 1.05 |
| **Max** | | 213.61 | 2.02 | 2.87 | 1295.42 | 1.20 | 2.12 | 1506.44 | 1.31 | 2.05 |
| | | $N_{SL}(N_i>10) = 22$ | | | $N_{SL}(N_i>5) = 48$ | | | $N_{SL}(N_i>3) = 64$ | | |
| μ | 4 | 135.66 | 1.10 | 1.54 | 395.51 | 0.77 | 1.38 | 671.34 | 0.50 | 1.34 |
| $\sigma_S$ | | 41.33 | 0.29 | 0.27 | 183.80 | 0.26 | 0.25 | 403.20 | 0.26 | 0.28 |
| **Min** | | 75.78 | 0.57 | 1.12 | 81.86 | 0.31 | 0.95 | 150.10 | 0.04 | 0.89 |
| **Max** | | 251.38 | 1.62 | 2.00 | 804.31 | 1.51 | 2.00 | 1566.48 | 1.10 | 1.96 |
| | | $N_{SL}(N_i>10) = 27$ | | | $N_{SL}(N_i>5) = 77$ | | | $N_{SL}(N_i>3) = 184$ | | |
| μ | 5 | 150.88 | 0.89 | 1.36 | 406.09 | 0.66 | 1.27 | 453.89 | 0.49 | 1.23 |
| $\sigma_S$ | | 69.94 | 0.23 | 0.31 | 184.59 | 0.21 | 0.26 | 174.28 | 0.22 | 0.26 |
| **Min** | | 51.16 | 0.60 | 0.88 | 99.05 | 0.15 | 0.72 | 52.05 | 0.08 | 0.72 |
| **Max** | | 255.16 | 1.62 | 2.02 | 801.94 | 1.16 | 1.91 | 963.48 | 1.06 | 1.88 |

**Table 4. *Spacing*, *CoV* and *V** statistical parameters. The Table reports the mean (μ), the standard deviation ($\sigma_S$), the minimum and maximum values of Spacing (*S*), *CoV* and *V** obtained from scanline statistical analyses.**

| | | **Cumulative length distributions** | | | **$P_{20}$ density distribution** | | | **$P_{21}$ intensity distribution** | | |
|---|---|---|---|---|---|---|---|---|---|---|
| | | $N_{(l>L)} = m \cdot L^{-\alpha}$ | | | $P_{20} = m \cdot s^{-\beta}$ | | | $P_{21} = m \cdot s^{-\delta}$ | | |
| | Set Type | *m* | α | $R^2$ | *m* | β | $R^2$ | *m* | δ | $R^2$ |
| Set 1 | A | 0.074 | 1.99 | 0.610 | 475 | 1.83 | 0.992 | 15.7 | 0.92 | 0.997 |
| Set 2 | A | 0.146 | 2.21 | 0.552 | 761 | 1.88 | 0.988 | 30.6 | 1.01 | 0.979 |
| Set 3 | A | 0.025 | 1.99 | 0.677 | 371 | 1.96 | 0.999 | 9.4 | 1.04 | 0.999 |
| Set 4 | B | 0.019 | 1.67 | 0.678 | 76 | 1.66 | 0.994 | 1.9 | 0.74 | 0.992 |
| Set 5 | B | 0.022 | 1.62 | 0.679 | 50 | 1.58 | 0.982 | 1.4 | 0.64 | 0.967 |
| Total | | 0.261 | 1.88 | 0.594 | 1187 | 1.77 | 0.989 | 37.6 | 0.86 | 0.986 |

**Table 5. Summary table reporting the power-law scaling and the values of the related parameters retrieved from the multiscale analysis of cumulative length distribution, lineament density and intensity.**

**Figures**

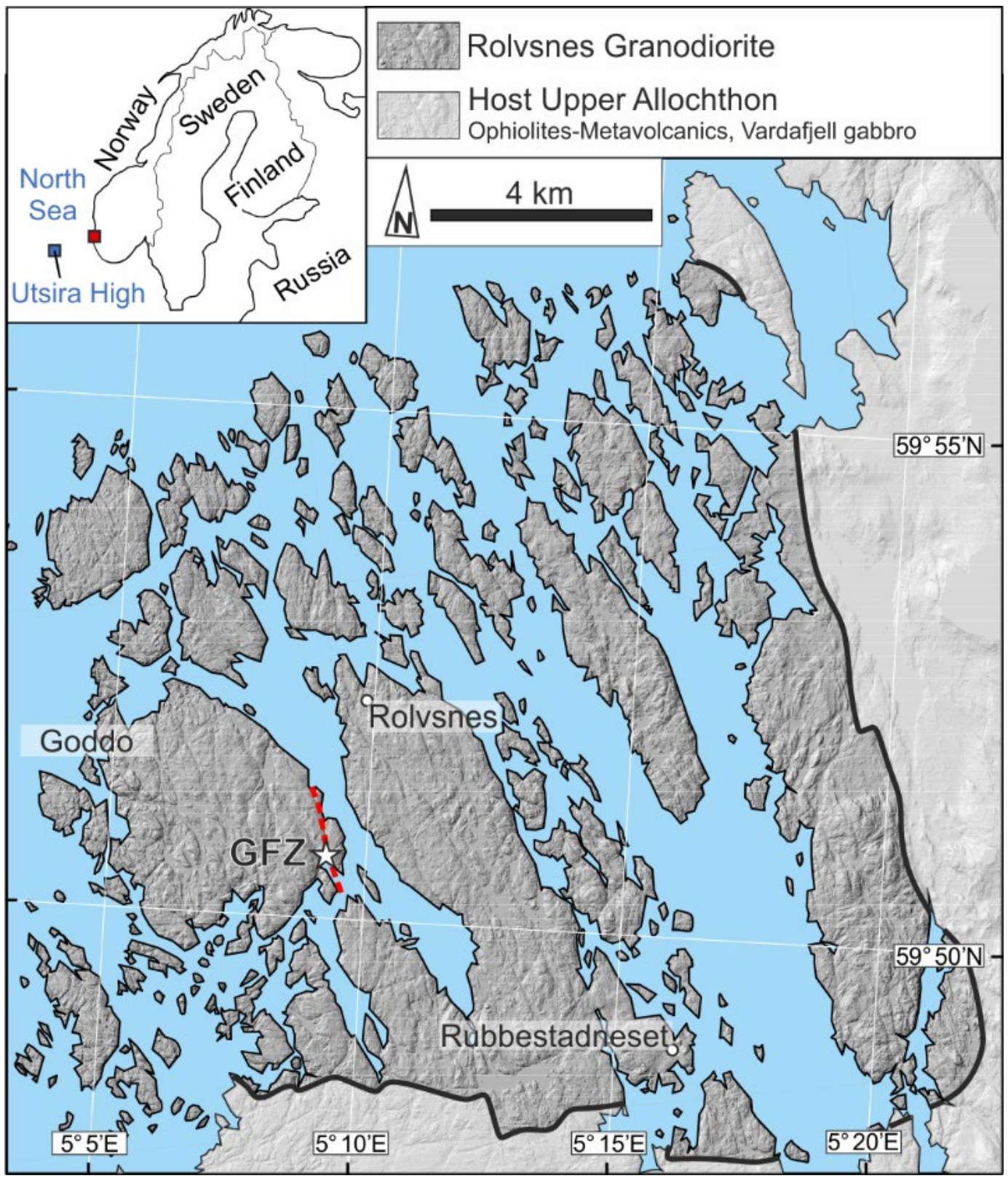

810 **Figure 1. Simplified geological map of part of the Bømlo Island centered on the Rolvsnes granodiorite overlaying the Digital Terrain Model obtained from high-resolution (1 m/pxl) LiDAR survey. The inset shows the location of the study area (red square) and the location of the Utsira High within the North Sea (blue square). The trace of the exposed GFZ is indicated by the red dashed line.**

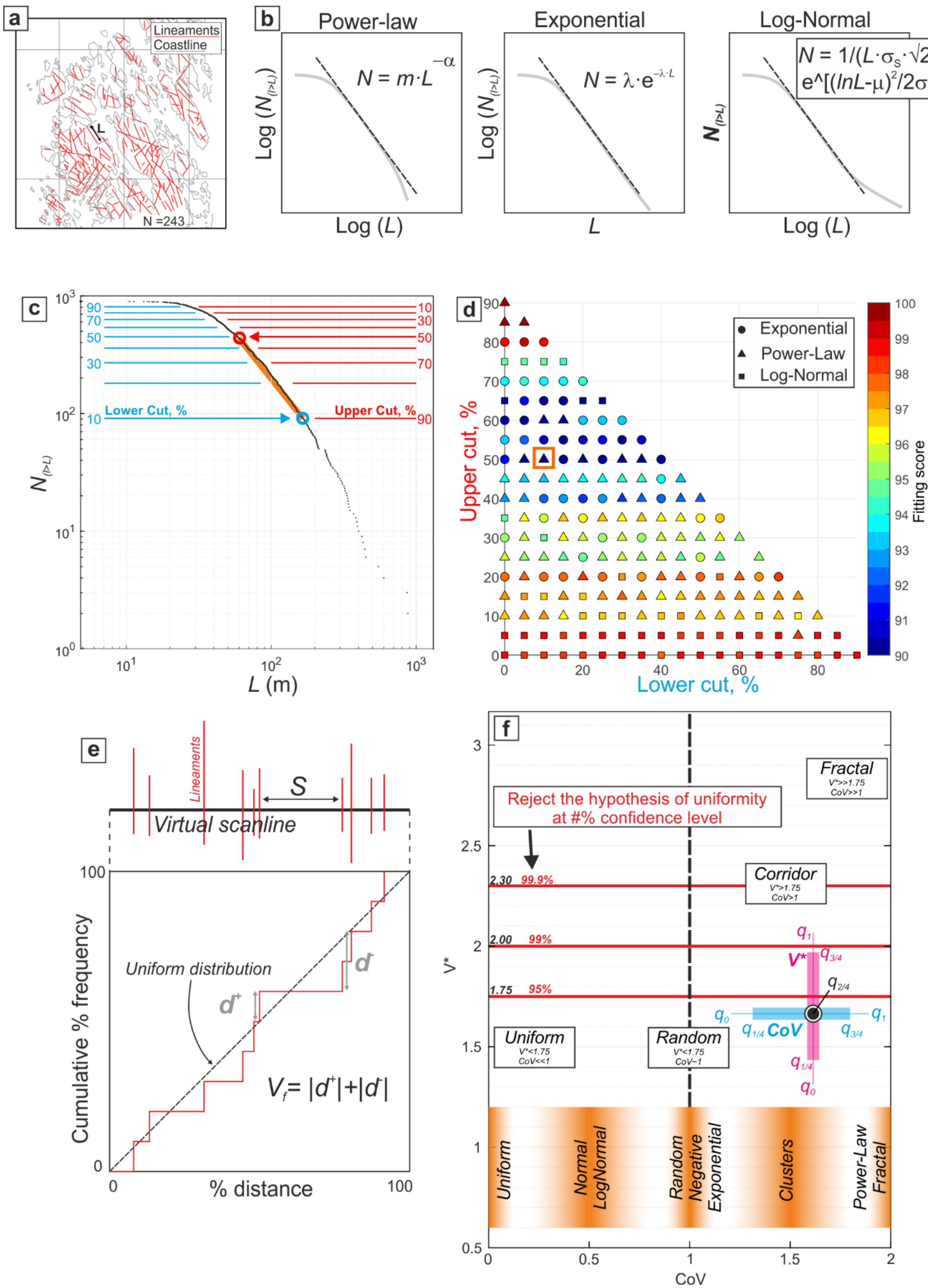

**Figure 2. Explanatory figure for the Method section. (a) Example of lineament map retrieved from the analysis of small scale DTMs. (b) Schematic representation of power-law, negative exponential and log-normal distributions, each of which defines a linear relationship between length *L* and cumulative number $N_{(l>L)}$ on a log-log, linear-log or log-linear diagram, respectively. (c) Example of cumulative length distribution, plotted on a log-log diagram, obtained from the analysis of lineament maps explaining graphically what the upper cut and lower cut values are. The blue and red circles represent the upper and lower cut values related to the checkerboard in (d). The orange segment represents the sub-domain of the cumulative distribution, included between the upper cut and lower cut bounds, fitted by the power-law relation identified by MLE-KS tests. (c) Example of checkerboard diagram. Each symbol (circle, triangle, square) represents a different fitting function, and each symbol is color-coded according to the fitting score yielded by the MLE-KS test for the portion of the cumulative distribution delimited by upper and lower cut values (plotted on the Y-and X-axis, respectively). The orange square represents the results of the MLE-KS tests performed on the distribution subdomain shown in (c). (e) Schematic representation of a virtual scanline and the related diagram showing the difference (*d* values) between the observed lineament distribution and a theoretical uniform (constant) distribution of spacings. (f) *CoV-V** diagram box-and-whiskers showing the expected ranges for uniform, random, clustered and fractal spacing distributions. The box-and-whiskers (light blue for *CoV*, cyan for *V**) report the values of the zeroth ($q_0$), first ($q_{1/4}$), third ($q_{3/4}$) and fourth ($q_1$) quartile of the distribution of *CoV* and *V** results. The central dot represents the median value of the results distribution (second quartile, $q_{2/4}$).**

830

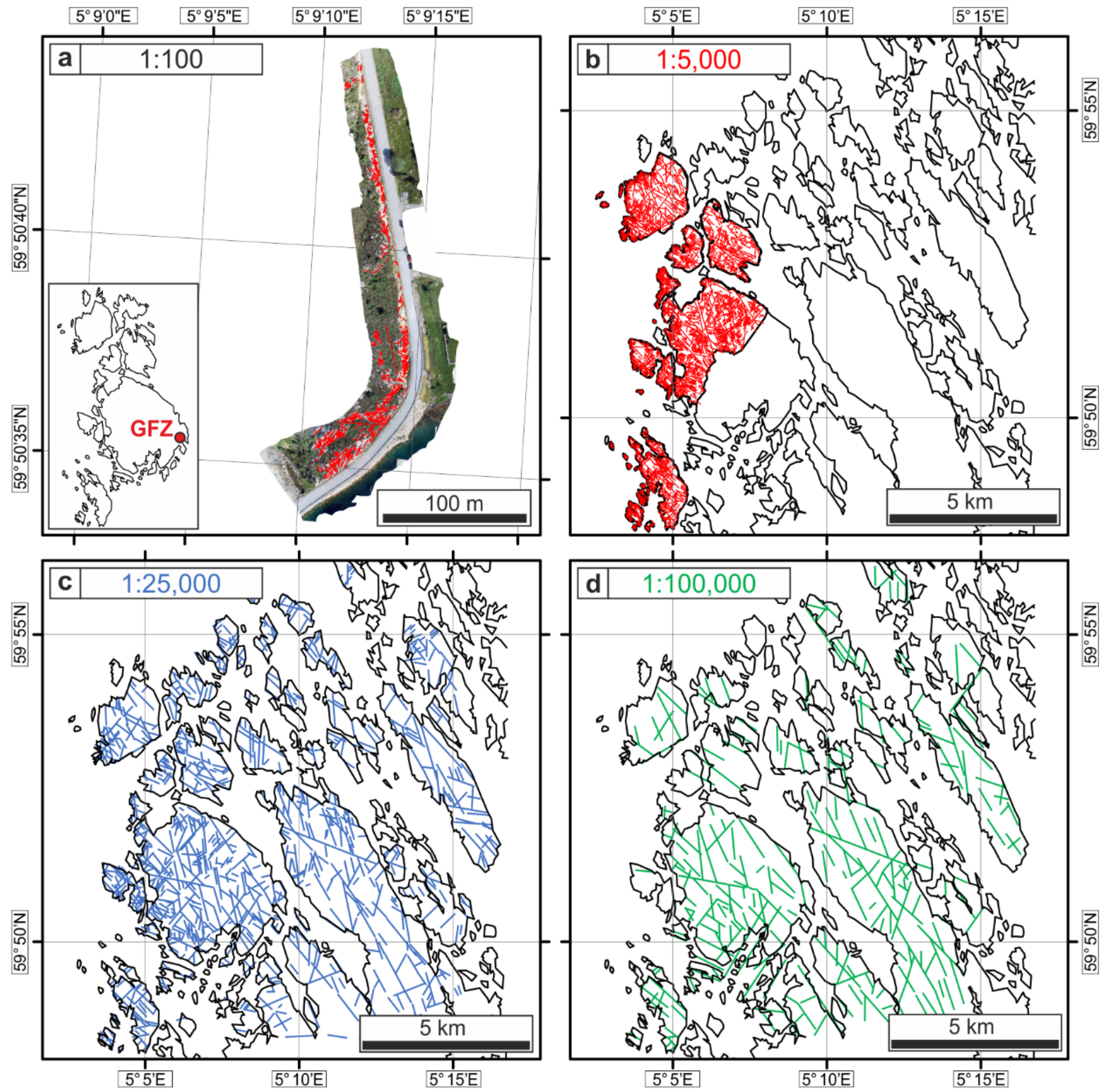

**Figure 3. Lineament Maps produced by manual lineament picking on outcrop orthophotos (a) and DTM from LiDAR surveys (b-c-d).**

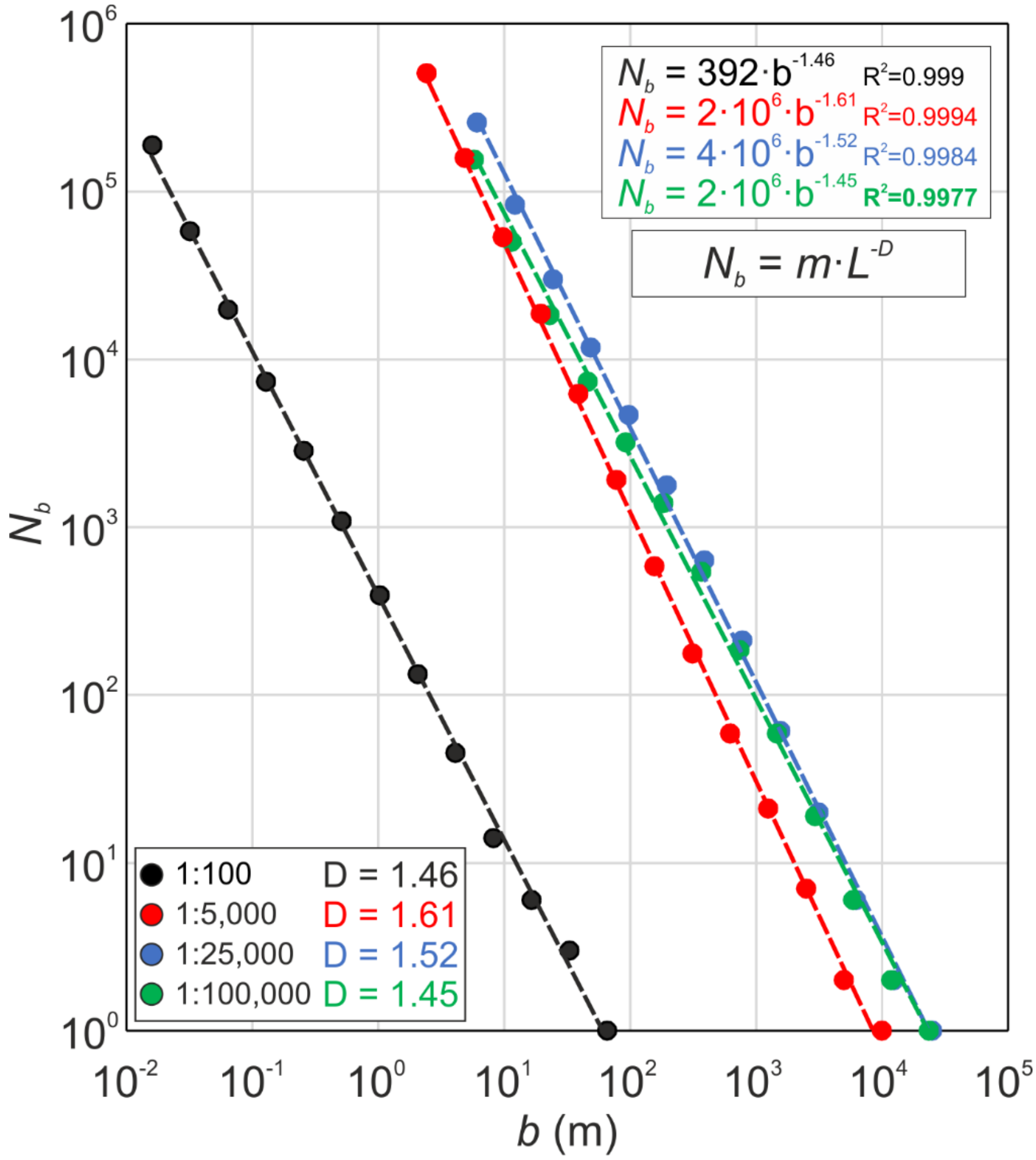

835 **Figure 4. Results of the box-counting method applied to the lineament maps of Fig. 3.**

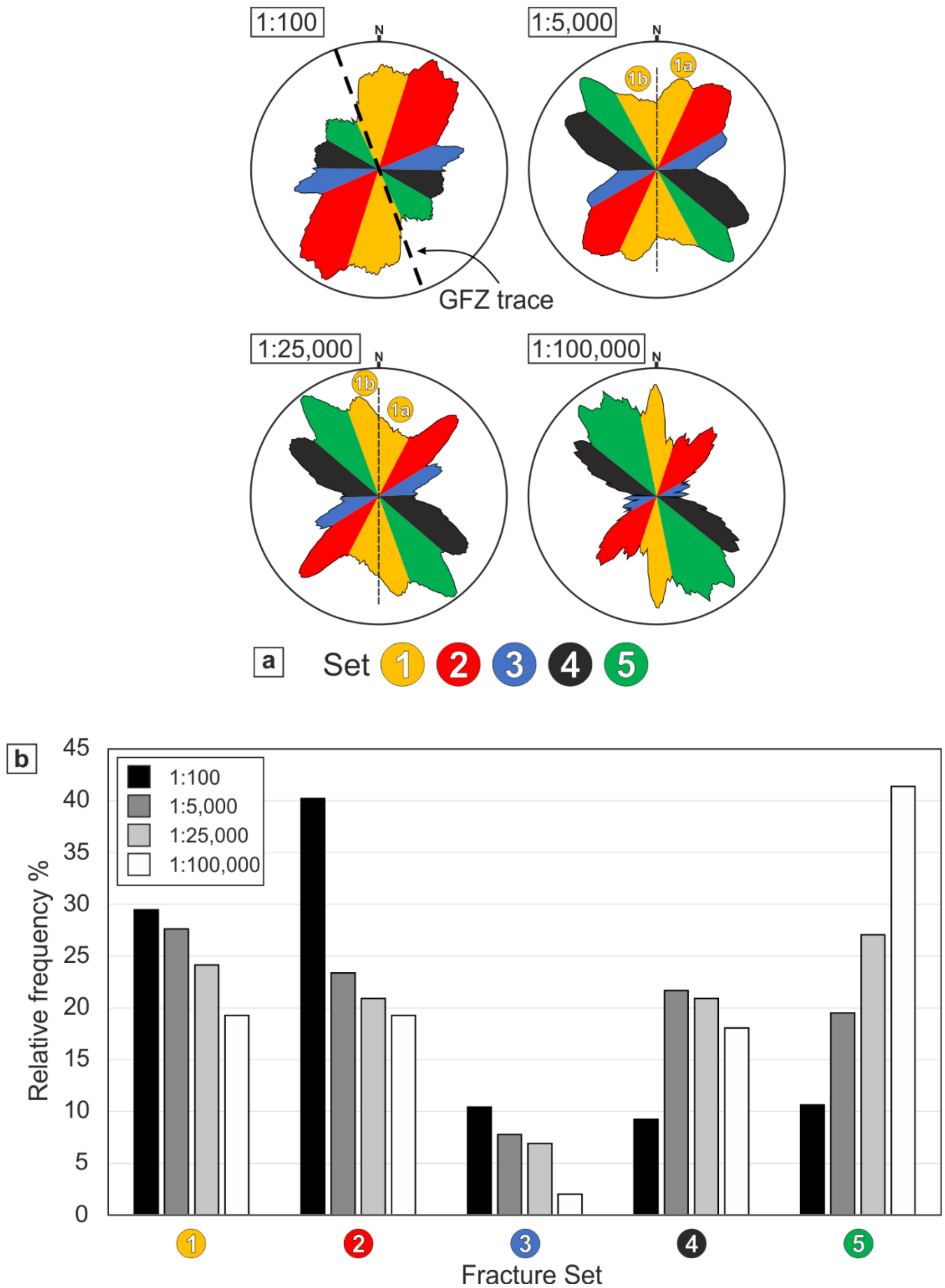

**Figure 5. Rose diagrams (a) and histograms of the relative frequencies (b) of the identified orientation sets at different scales of observation.**

840

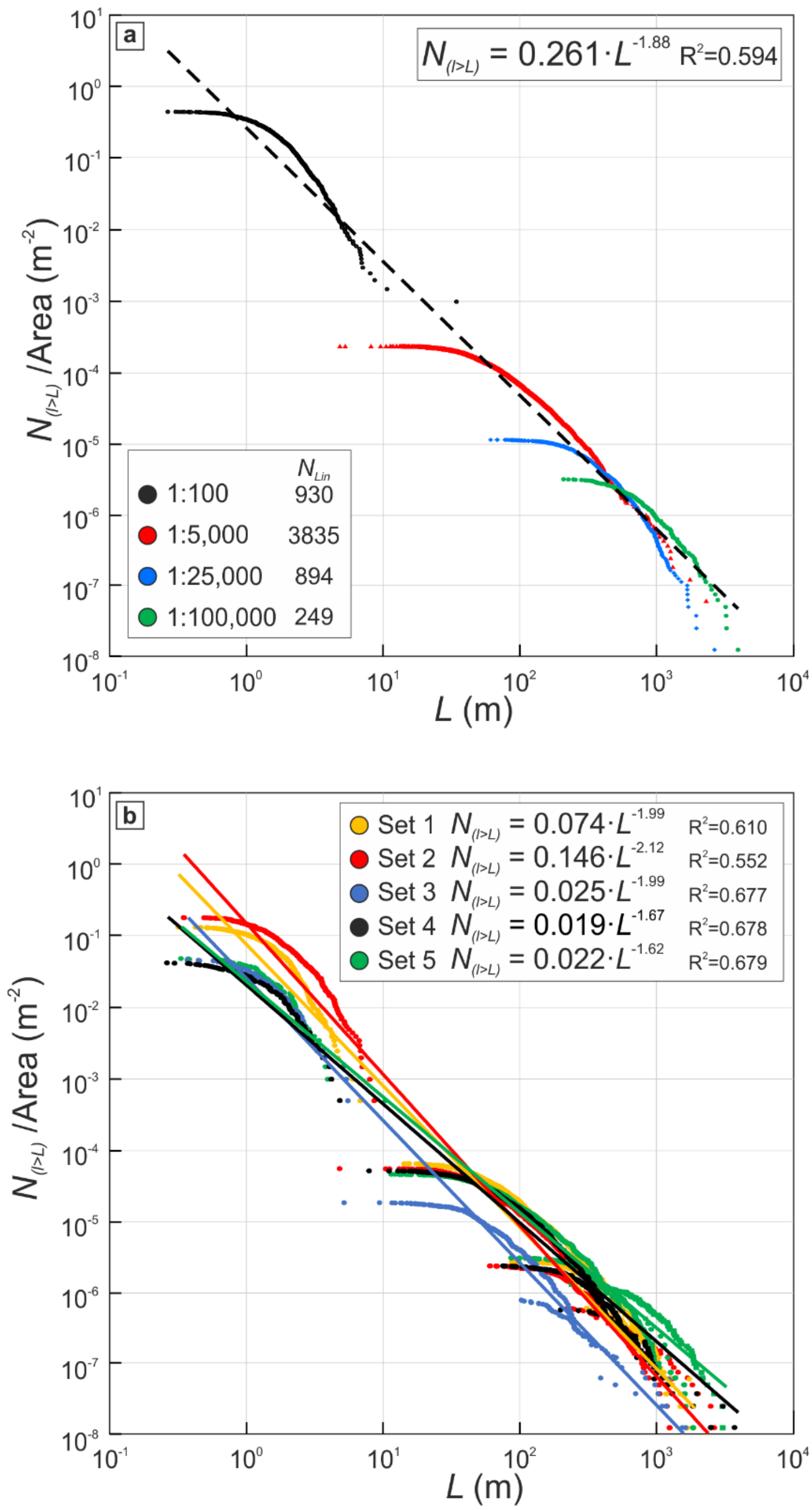

**Figure 6. (a) Log-log diagram of lineament length *L* vs. cumulative number of lineaments $N_{(l>L)}$ per unit area showing the cumulative length distribution for the whole lineament maps reported in Fig. 3. (b) Log-log diagram as above showing the cumulative length distribution for each orientation set.**

845

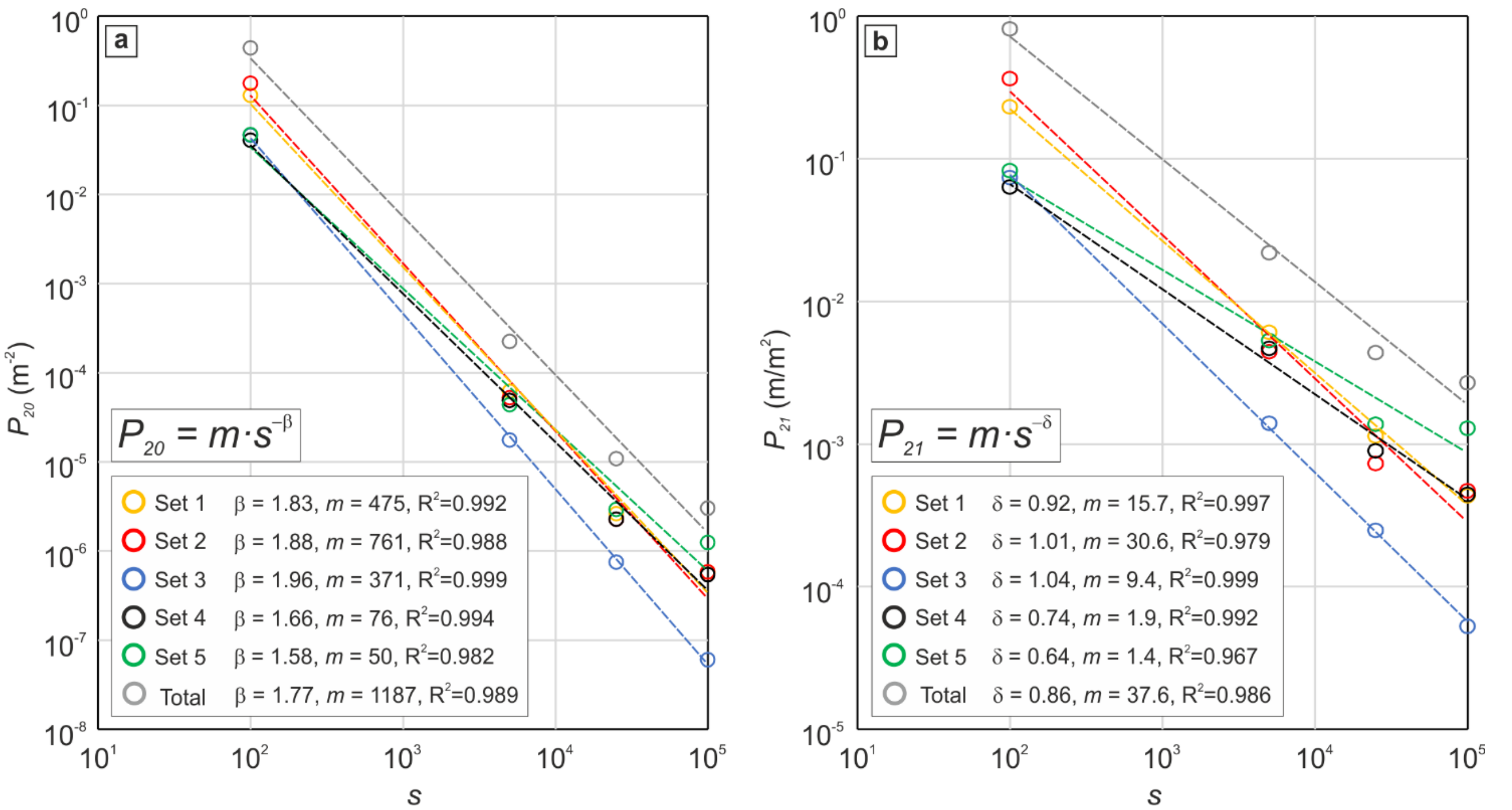

**Figure 7. Lineament density ($P_{20}$) and intensity ($P_{21}$) variation across scales for each orientation set (Set 1 to 5) and for the entire lineament network (Total).**

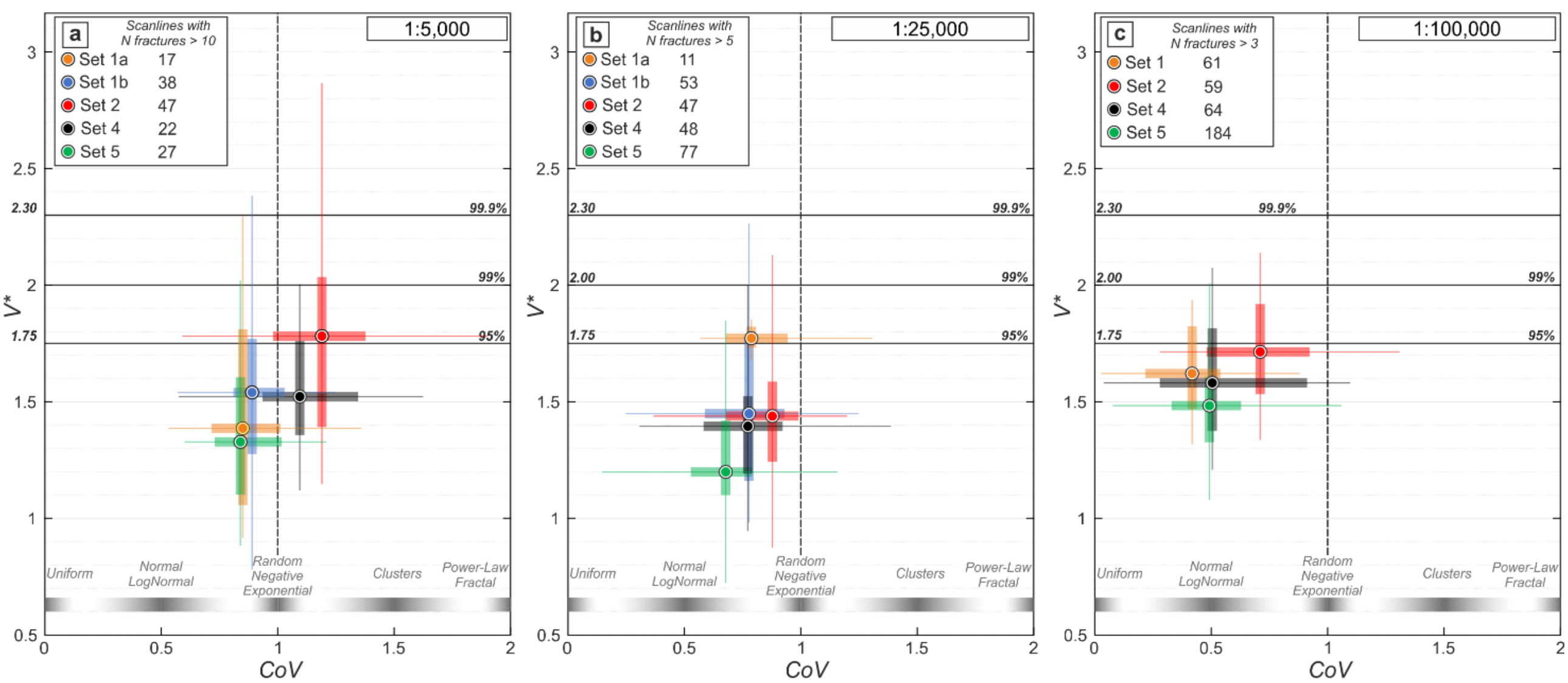

**Figure 8. Box-and-whiskers plot reporting *CoV-V** values quantifying the spatial organization of the orientation sets identified in the lineament map at (a) 1:5,000; (b) 1:25,000; and (c) 1:100,000 scale.**

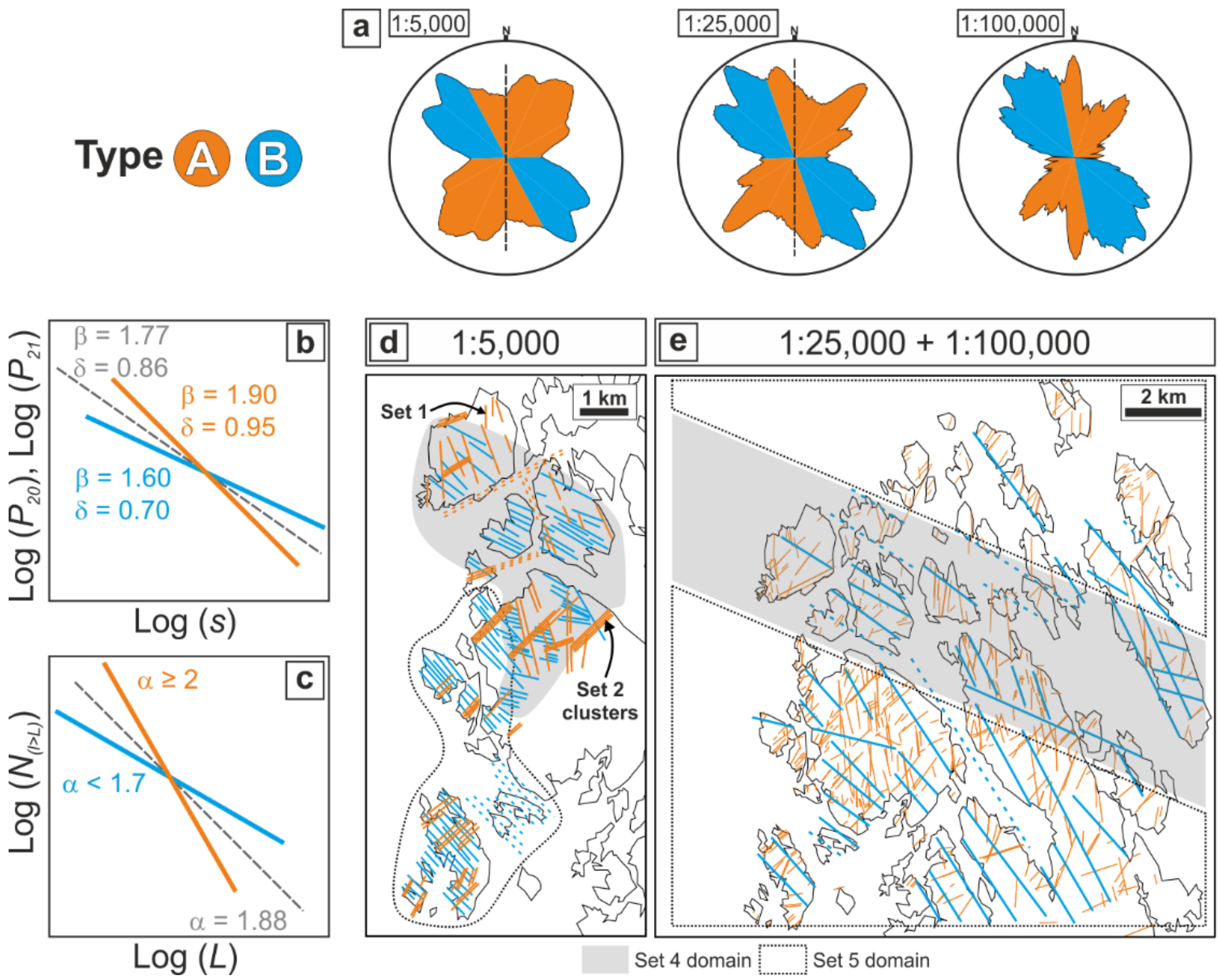

**Figure 9. Schematic summary of the results and interpretations for the Rolvsnes granodiorite case study. (a) Rose diagrams of the orientation of lineaments at different scales (1:5,000; 1:25,000; 1:100,000) with the classification into Type A and B lineaments. (b) Schematic log-log diagram showing the observed general trends of $P_{20}$ and $P_{21}$ variations with scale. The values of β and δ exponents are reported for the entire lineament network (grey dashed line), Type A (orange line) and Type B (light blue line) lineaments. (c) Schematic log-log diagram showing the observed general scaling laws retrieved for the cumulative length distributions. The values of the exponent α are reported for the entire lineament network (grey dashed line), Type A (orange line) and Type B (light blue line) lineaments. (d) Schematic representation of the lineament distribution at 1:5,000 scale. The reported lineaments are redrawn from the 1:5,000 lineament map and represent the spatial organization observed within the Rolvsnes granodiorite. (e) Schematic representation of the lineament distribution at 1:25,000–1:100,000 scales. The reported lineaments are redrawn from the 1:25,000–1:100,000 lineament maps and represent the spatial organization mapped on the Rolvsnes granodiorite. Note the clustered organization of Set 2 lineaments and the two domains (highlighted by transparent grey and dashed areas) where Set 4 and Set 5 lineaments are dominant, respectively.**