# Peer review of "Multiscale lineament analysis and permeability heterogeneity of fractured crystalline basement blocks"

_EGUsphere, 2022_

## Author Response (AR1)

We would like to thank the Reviewers for their constructive comments and helpful advice. We have addressed all the points raised by the Reviewers. In the followings we present the replies (reported in *Italic font*) to the Reviewers' comments (reported here in gray).

Reviewer #1

Review of the manuscript "Multiscale lineament analysis and permeability heterogeneity of fractured crystalline basement blocks" by Alberto Ceccato and coauthors.

The paper presents a multiscale geometrical analysis of lineaments within a granodiorite pluton in the basement of the southwestern Norway, based on the interpretation of a DTM (Digital Terrain Model) at the 1:100000, 1: 25000 and 1:5000 scale, and an UAV-mapped zone at the 1:100 scale.

The paper is well written, and figures are adequately illustrating the results. The proposed methodology combines several methods used in previous studies, including box-counting on lineament maps, fracture orientations, fracture length cumulative distributions, fracture spacing distribution along virtual scanlines, fracture intensity and density at multiple scales. The analytical procedure is repeated also on fractures subdivided by set, allowing to define a hierarchical organization of the fracture network, with fractures controlling the network at the regional scale (type B fractures) and the other sets (type A fractures) having effects at smaller scales. These data are then discussed on the basis of detailed field and petrophysical data published by the same authors, and a conceptual model of the fracture permeability of the Rolvsnes granodiorite is proposed.

The paper is well suited to be published in this journal. I have some concerns about the analytical design and methodological approach, as discussed in the general comments below. Due to these issues, the paper needs some revisions before publication.

General comments

In the study area, the outcrops are fragmented into many islands, which in turn are likely to have a fractal size distribution. The fragmentation of the outcrops probably influences the fractal dimension measured from box counting analysis, and the length distribution of the lineaments (i.e., the maximum length is controlled by the size of the islands). Have you tested whether and how the geometry of the outcrops influences the fractal geometry of the network? Specify the area considered for the box counting analysis and discuss the possible resulting biases.

**Reply.** *We agree with the Reviewer and indeed we have now performed further box-counting analyses to evaluate the effect of such fragmentation.*

*We have now analyzed with the box-counting technique several additional selected areas completely contained within the islands and not affected by the fractal character of the exposed*

*land surface of the islands. We have modified accordingly Figure 4, which now includes the new data as box-and-whiskers plots. The new results are, additionally, also integrated in the main text. The newly analyzed sub-areas are reported in the Supplementary material and the result dataset is reported in the online repository.*

*The obtained results indicate a weak variation trend of the fractal dimension across scales, which, in our opinion, is consistent with the other variation trends observed for the lineament properties such that this strengthens our initial discussions and subdivision of the lineament sets in local- and regional-dominant orientation sets.*

*Also, the results are still comparable with the fractal dimensions obtained from other studies where there is no fragmentation of the analyzed surface.*

The shape of the islands is often concave (see for instance the map at the 1:5000 scale). 2D quantifications of fracture networks is generally done on convex polygons (scan areas), because otherwise the same line might be counted twice, and the effects of censoring are enhanced. Did the authors consider this effect on the length distribution?

**Reply.** *Yes, we discuss this topic in Section 5.2.4, also citing Cao & Lei (2018). However, given that the lineaments are partially covered by the sea, one cannot conclude whether the colinear lineaments cropping out in different parts of the concave islands are just segments of the very same lineament or, instead, two distinct segments.*

There is a large gap between 1:5000 and 1:100 scales, and the area close to the Goddo Fault zone mapped by UAV survey is L-shaped and very thin (50 *200 m). I would be careful in combining the lineament length cumulative distribution in the UAV map with those of the maps at a smaller scale. This results in a significant range of lineament lengths not being covered in Fig. 6.

**Reply.** *See reply below.*

The R2 for the linear regression are very low in Fig. 6 (from 0.552 to 0.679), and this makes the regressions meaningless, especially in Log-Log space. In Log-Log space, a R2 of at least 0.9 is recommended to fit a power law distribution.

**Reply.** *Regarding the statistical significance of low $R^2$ values in Fig. 6, we would like to remark that if we followed the $R^2 > 0.9$ approach, we should then reject most published results on distribution fitting of geological parameters. In the considered literature, indeed very few papers calculate and show $R^2$ for LSR fitting; fitting of length distributions is instead usually made manually/qualitatively. The review and analysis of similar datasets (e.g., particle size distribution) the literature with statistical and quantitative methods has revealed that most of the scaling-laws derived by qualitative/manual fitting in the literature are very weak from a statistical point of view and many conclusions would actually need reconsideration (e.g., Phillips and Williams, 2019; https://doi.org/10.1016/j.epsl.2020.116635).*

*Our $R^2$ values are low because the LSR fitting is made on the whole distributions, without any manipulation or elaboration of the datasets (for example, also considering the part of each distribution affected by censoring and truncation, as already explained in the main text). This issue is addressed and discussed at lines 435-443 of the submitted manuscript.*

Moreover, the shape of the surveyed area makes it difficult to intercept lineaments striking NS to NW, which might be one of the reasons for shortage of these orientations at the 1:100 scale in Fig. 5. The biasing effects of the scale gap and outcrop shape should be discussed with more detail in section 5.2.

**Reply.** *We thank the Reviewer for the comment. We have modified accordingly our discussion paragraph (former Lines 394-400).*

The lack of a unique fractal distribution fitting the multiscale fracture sets implies that the network is not self-similar.

**Reply.** *We have now modified the data presentation and discussion of the fractal dimension Indeed, we now demonstrate that these variation in fractal dimension across scale is related to the variation in relative occurrence of fracture orientation sets, which show scale-invariant geometrical properties. In any case, the fractal dimensions of entire lineament maps are still similar, within uncertainties (±0.15).*

This finding is coherent with the fact that the granodiorite experienced multiple stages of brittle deformation, but it doesn't seem the most common spatial arrangement found in similar settings. In my opinion, the discussion lacks a comparison with previous multiscale studies of fracture networks, which would allow to appreciate the pros of the suggested methodological approach and of the integration of statistical analysis with field geology.

**Reply.** *We thank the Reviewer for the comment; however, we think that we already address this point in the first paragraph of the discussion section (Section 5.1. Lines 319-328), where we do say and comment on the fact that the retrieved scaling parameters for the Rolvsnes granodiorite are rather similar to those obtained from the analyses of other crystalline basement units elsewhere.*

Line by line comments

Lines 12-14: check this statement, which contradicts the following sentence (is there a scale-invariant spatial distribution or not?). See also general comment 1.

**Reply.** *The spatial "distribution", quantified by the fractal dimension D, refers to the spatial occupancy of lineaments and it is different from the spatial "organization" which refers to the distribution of spacing of fractures along scanlines (Bonnet et al., 2001; https://doi.org/10.1029/1999RG000074). There might be some correlation between the two but demonstrating this goes beyond the scope of the present paper. Therefore, the two sentences do not contradict each other.*

Line 14: the symbol is missing

**Reply.** *Text modified.*

Line 49 – Paragraph 1.2 doesn't seem necessary; the structure of the paper is quite conventional. Consider removing this paragraph, which basically anticipates the information given in a more detailed way in the Geological setting and Methods sections.

***Reply.*** *We prefer to keep the paragraph as it represents a brief guide for the Reader to follow the logical progression of the paper.*

Lines 98 – 106: it would be useful to introduce in Fig. 1 a sketch shoving the orientation of the structures (faults and fractures) associated with each tectonic stage. Here, you could add some information about the dip angles: are fractures all subvertical, therefore justifying your analytical approach in map view? Are there oblique sets?

***Reply.*** *We prefer to keep the figure as simple as possible, without adding any further detail the utility of which we consider arguable in the present paper. We address the reader to other published papers where the tectonic deformation sequence and related structures are described in great detail. The lineaments are mostly subvertical (see Scheiber and Viola, 2018).*

Lines 114 – 119: the relationships with the offshore reservoir could be mentioned in the introduction.

***Reply.*** *The relationship with the offshore reservoir is a side-topic of the paper, relevant for regional geology, and we therefore prefer to keep the introduction focused on the description of the general implications of our study. The Reader is addresses to more detailed papers on the regional geology (Scheiber and Viola, 2018; Viola et al., 2016; Ceccato et al., 2021a,b,).*

Line 120: delete "Materials and"

***Reply.*** *Text modified accordingly.*

Lines 159 – 199: I'm a little confused about the procedure of fitting single-scale and multiscale cumulative distributions. For single scale data, you test three possible distributions and score the best one fitting truncated or non-truncated data. For multiscale analysis, you assume that all distributions are fitted by a power law, without truncation. I think that you should clarify your reasoning here. You might try to fit the multiscale power law distribution by using truncated data (see for instance Chabani et a., 2021).

***Reply.***

*How Chabani et al. (2021) computed the multiscale fitting is not clear to us and it would appear that it is not explained in enough detail, such that their method is not reproducible.*

*Our paper demonstrates that several subdomains (defined by different Upper and Lower cuts) of a distribution exist where the same fitting function (e.g., power law) may apply, scoring similar ranking values in the KS tests. Each of these subdomains are characterized by different fitting parameters. As a consequence, we would not know how to define which is the right subdomain and function parameters?*

*As already discussed, in most cases truncated distributions report even less than 50% of the whole dataset, implying that their statistical significance and representativeness are rather arguable.*

*Rather than accurately quantifying the scaling parameters, our paper aims to highlight the weaknesses and uncertainties that exist in these sorts of analyses even when very robust statistical methods are applied. To clarify this aspect, we have modified the text and implemented this comment in the Introduction and Discussion sections.*

Lines 229 – 230 and 237 – 238: 3 intersections per scan line are very few for these considerations – if your dataset does not fulfil the requirements for a statistically meaningful analysis, it's better not to perform that analysis.

**Reply.** *We think that, even though the number of intersections for each scanline is indeed limited, the overall number of analyzed scanlines might be statistically significant. Therefore, we prefer to keep the presented data.*

Lines 242 – 251: this is merely a list of the number of picked lineaments – could you add some qualitative description about length, intersections, orientations? The lack exposure below the sea level can be introduced here.

**Reply.** *The length, intersections, orientations are described in the following sections of the Result description.*

Line 252: already commented, consider the effects of the fragmented exposures.

**Reply.** *See reply above.*

Line 275: remove reference to Dichiarante et al., 2020

**Reply.** *Done.*

Lines 271 and following lines: here, you could mention which of the datasets meet the minimum number of 200 fractures (now only in the supplementary material)

**Reply.** *The information is already reported at the beginning of section 4.1.*

Lines 282 – 287: as commented above, the multiscale cumulative length distribution is not that robust, because there is a significant range of fracture lengths which is not covered by data (the tens of metres range), and both the large scale fracture lengths (1:100) and the small scale fracture lengths (1:100000) are not fit by a power law (if I understood well Table 3). To overcome this last point, you might consider only the part of the distributions having a power-law distribution for the multiscale fit. Lower and upper cut of the power-law distributions can be obtained statistically with the MLE method of Clauset et al (2009) (https://aaronclauset.github.io/powerlaws/).

**Reply.** *We would like to remark that most of the published data about power-law fitting of multiscale distribution are usually retrieved by qualitative, manual fitting, and thus they completely lack any quantification of their intrinsic "statistical robustness".*

*The purpose of doing multiscale comparisons is indeed that of retrieving a mathematical function describing the portions of a general distribution which are missing from the analyzed dataset.*

*Yes, it is true indeed, the small and large scale are not fitted by a power-law function, yet they plot along a general power-law trend.*

Line 298: what does it mean a decreasing trend?

**Reply.** *Text modified.*

Line 300: ";" > ","

***Reply.*** *Text modified.*

Lines 321-322: see comment above about cumulative length distribution.

***Reply.*** *See reply above about cumulative length distribution.*

Line 327: and what about P21?

***Reply.*** *We are not aware of any paper reporting the power law exponent of the multiscale $P_{21}$ variation, thus we cannot compare it to any other published dataset.*

Lines 339 – 340: it is probably Fig. 6 and not 5

***Reply.*** *Text modified accordingly.*

Line 347: check the sentence

***Reply.*** *Text modified.*

Line 373 and following lines: as suggested in the general comments, the possible biases due to the outcrop shapes are not considered. I suggest evaluating them as well.

***Reply.*** *We have modified the section including the Reviewer's comment.*

**Reviewer #2**

The authors present a study of the fracture patterns in onshore exposures of the Rolvsnes granodiorite. The methodology adopted is state-of-the art and includes some novel features that could eb highlighted further in the text. The results are well described and illustrated in figures including the supplementary files. The discussion of biases in the data is useful and the authors conclusion that they have identified a multiscale fracture hierarchy is justified. The study will be interesting to those working on fracture reservoirs generally and adds to our growing knowledge of fractured basement systems.

A few suggestions and comments on specific items in the manuscript.

Line 79-80. I suggest to moving this paragraph up to the start of this section because it a) discusses previous work and b) gives a statement of the problem and need for further constraint. This reorganisation would mean that this section on 'Structure of the paper' finishes with the lines that are now 76-78 which highlights the contribution in this study. Furthermore I suggest combining the 2 paragraphs in lines 70-78 together into 1.

**Reply.** *Text modified accordingly.*

Line 115-118 - So what? Explain why are you telling the reader this. Can the study of the Rolvsnes basement tell you something about the Utsira high? If so what can it tell us?

**Reply.** *We have rephrased the sentence in order to clarify the relationship between the Rolvsnes granodiorite and the Utsira High and why we are explaining this here.*

Lines 150-155 - One of the criticisms of box counting is that the results can be biased if the exposure boundaries that are different from the sample boxes. Reliable results can be obtained where the boxes are entirely within the mapped data - How did you take this into account given the exposures in the study area are a series of islands? Did you test to see if the results are reflecting the shape of the islands rather than the fractures within them?

**Reply.** *See reply to the comment of Reviewer #1.*

Line 239. It was claimed in the abstract that the workflow presented is novel. Could the authors explain here or at the start of this section what is novel here - as opposed to 'State of the Art'. To me the workflow represents current best practice but I struggle to see where the novelty lies.

**Reply.** *We think that the integration of such "state-of-the-art" methods constitutes a novel approach to the analysis of lineament maps.*

Line 373 spelling of 'and' in subtitle.

**Reply.** *Text modified accordingly.*

Line 398 Have you tried plotting rose diagrams that are length weighted? This can bring out dominant trends.

**Reply.** *No, we didn't.*

Figures and Figure captions

Line 810 - What does GFZ stand for? - reader shouldn't have to refer to the text for the abbreviations in the figures.

**Reply.** *Text modified to explain the acronyms.*

Line 825 - What do the abbreviations CoV and V* stand for? Reader shouldn't have to refer to the text.

**Reply.** *Text modified to explain the acronyms.*

Figure 9. it would be useful here to add a reminder of the set numbers - the text (c. Line 365) discusses the fractures by set number but only types appear on the Figure

**Reply.** *Figure modified accordingly.*

---

## Author Response (AR2)

Review of the revised manuscript "Multiscale lineament analysis and permeability heterogeneity of fractured crystalline basement blocks" by Alberto Ceccato and coauthors.

The authors fairly addressed the reviewers' observations on the submitted ms, by presenting some additional analyses and modifying the main text in several parts. The answers to reviewer comments are complete and well supported. I still have a couple of comments, which can be easily addressed with some minor revisions. Line and paragraph numbers refer to the tracked-changes manuscript.

1. The analysis of the effects of the outcrop shape on the fractal dimensions of the network is introduced in paragraph 4.2 and Fig. 4, but these data are not discussed. Please, discuss the effects (if any) of outcrop fragmentation - this is likely to be a further bias in addition to those presented in paragraph 5.2.

**Reply.** *Text modified accordingly. "Even if there is a small variability in the fractal dimension D values, none of the measure fractal dimensions, either from small sub-areas or entire lineament maps, are similar to the fractal dimension D = 1.77 retrieved from the areal occupancy of the exposed land surface of the Bømlo Island archipelago. The relationship between fractal dimensions of exposed land surface and fracture maps remains to be understood and would deserve further analyses which go beyond the scope of the present paper."*

2. As already commented in my previous review, I think that the main limitation in the multiscale analysis of fracture lengths is that there is a significant scale gap between the 1:5000 and 1:100 scales. If you look at Fig. 6, the three datasets collected at small scale collapse reasonably well on a line, while the 1:100 data doesn't lay on the same line (neglecting the fact that the length cumulative distribution at the 1:100 scale is best fitted by a negative exponential, line 317). This tendency is even more evident by subdividing the data by set (part b). The lack of fractures in the range of the tens of meters of length does not allow to estimate at which scale this variation in slope occurs, and to test more sophisticated geometrical models (e.g., multifractals). I suggest making it clear from the beginning that you are dealing with an incomplete dataset, which is quite a common condition, unless you have exceptionally wide outcrops.

**Reply.** *We have now modified the introduction section accordingly: "This multiscale and statistical approach tries to overcome the natural bias of lineament maps retrieved from remote sensing of natural outcrops, which are inherently incomplete due to partial exposure, resolution and analytical biases."*

3. I'm aware that everybody normalizes cumulative distributions by area. However, this normalization assumes that Ntot/A = P20 is constant across scales, which is not true for your data (Fig. 7, § 4.5). Since you reconstruct a density model (with β = 1.77 for the whole set), why don't you use it to normalize the length distributions? See for instance Bour et al., 2002, § 5.2. The paper is cited in your reference list. Or at least you can discuss the variation in density with scale as an analytical bias possibly affecting the determination of the multiscale length distribution.

**Reply.** *This would be an interesting analysis that actually goes beyond the scope of the present paper. In addition, the change in density could be a couple effect of both change in resolution and increasing/decreasing actual abundance of lineaments across the scales of observation.*

Line 11: fracture or lineament? Please, choose one. I suggest using lineament here

**Reply.** *Text modified accordingly.*

Line 14 – 15: relative abundance is not clear to me, pleas rephrase

**Reply.** *We think that "relative abundance" is the correct term here, because it refers to the percent composition of one set of lineament relative to the total number of lineaments in the maps.*

Line 25: -18 superscript

***Reply.*** *Text modified accordingly.*

Line 386: delete "proper"

***Reply.*** *Text modified accordingly.*

Line 884: courtesy

***Reply.*** *Text modified accordingly.*